# Multimodal Nested Learning for Decoupled and Coordinated Optimization

**Yanglin Feng**[1]  **Yang Qin**[1]  **Dezhong Peng**[1,2]  **Rui Wang**[1]  **Xiaomin Song**[3]  **Peng Hu**[1]

## Abstract

Multimodal learning aims to integrate multi-sensor data to exploit their complementary information, embracing a more comprehensive real-world perception and understanding. However, heterogeneous discrepancies across modalities consistently trigger imbalanced multimodal optimization, restricting the joint learning performance. Although existing methods mitigate this issue through optimization modulation and conflict alleviation, they still suffer from entangled optimization and uniform learning pace in conventional monolithic frameworks, limiting the effectiveness of multimodal learning. To address this issue, we propose a novel **M**ulti**mo**dal **Nest**ed Learning Framework (MoNet), which reformulates the monolithic framework into nested sub-processes, decoupling and coordinating multimodal learning. To achieve this, we present a Decoupled Multimodal Stable Memory block (DMSM) as the outermost nested level, which decouples multimodal learning into independent optimization streams for semantic exploitation across modalities. Additionally, we develop an Adaptive Multimodal Coordinated Fusion block (AMCF), which constitutes the inner nested level. It attempts to coordinate multimodal information integration across multi-timescale nested memories, balancing multimodal fusion. Extensive experimental results on eight datasets across three tasks demonstrate the superiority of MoNet. Code is available at https://github.com/Yangl1nFeng/MoNet.

[1]College of Computer Science, Sichuan University, Chengdu, China [2]Tianfu Jincheng Laboratory, Chengdu, China [3]Sichuan National Innovation New Vision UHD Video Technology Co., Ltd., Chengdu, China. Correspondence to: Peng Hu <penghu.ml@gmail.com>.

*Proceedings of the 43$^{rd}$ International Conference on Machine Learning*, Seoul, South Korea. PMLR 306, 2026. Copyright 2026 by the author(s).

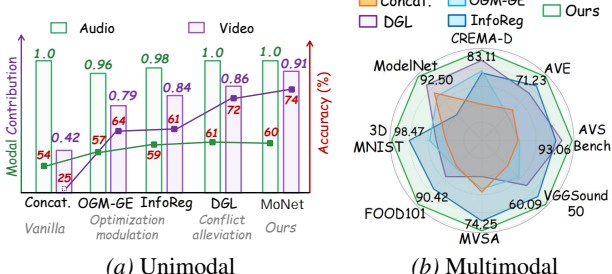

*Figure 1.* Comparison between multimodal learning methods (*i.e.*, Concat., OGM-GE, InfoReg, and DGL) and our proposed MoNet. To be specific, (a) compares unimodal contribution (Peng et al., 2022) (*i.e.*, bars) and performance (*i.e.*, lines) on the CREMA-D dataset. Specifically, the former measures how much a single modality supports the classification objective, whereas the latter reports the accuracy with only unimodal inputs. Their computational details are provided in our Appendix. (b) depicts a comparison of multimodal fusion recognition accuracy on eight datasets.

## 1. Introduction

Humans perceive the world by integrating information from multiple senses, such as vision, audition, and language, to form a coherent understanding of the real-world environment (Stein & Meredith, 1993). Inspired by this process, multimodal learning endeavors to emulate human perception by jointly modeling and coordinating heterogeneous data from different modalities, enabling comprehensive cognition and reasoning. By leveraging the complementary information across modalities, multimodal learning can effectively mitigate the inherent limitations of unimodal perception and has demonstrated strong practical performance in audio-visual recognition (Ma et al., 2025) and image-text classification (Xu et al., 2025; Lan et al., 2025) tasks.

However, due to the substantial heterogeneity across modalities, multimodal learning often fails to deliver the expected performance gains and could even underperform unimodal learning in practice. Existing studies (Peng et al., 2022; Huang et al., 2022) attribute this issue to imbalanced optimization across modalities, where a strong modality tends to dominate the training process, inhibiting effective extraction of task-relevant semantics from other modalities. Intuitive solutions (*e.g.*, OGM-GE (Peng et al., 2022) and InfoReg (Huang et al., 2025)) are to modulate the optimization pace of each modality to suppress strong modalities and

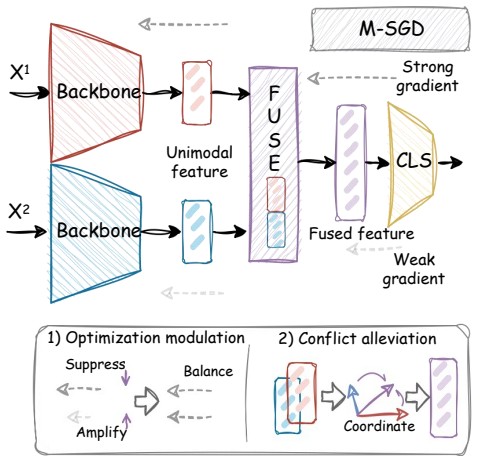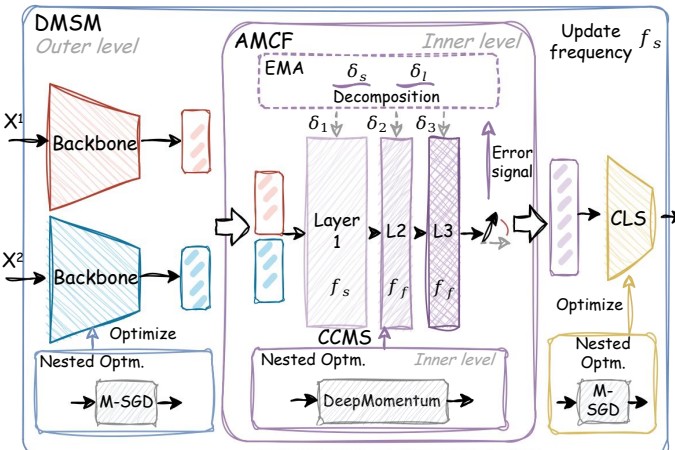

*Figure 2.* Overview of the conventional multimodal learning framework and our proposed framework. Specifically, the left shows the conventional framework and two representative approaches for handling modality imbalance. The right depicts the pipeline of our Multimodal Nested Learning Framework (MoNet). The outer layer is a Decoupled Multimodal Stable Memory block (DMSM), designed to separately exploit modality-specific information for classification, optimized by independent momentum SGD optimizers (M-SGD). The inner layer is an Adaptive Multimodal Coordinated Fusion block (AMCF), which utilizes a multi-timescale Coordinated Continuum Memory System (CCMS) for adaptive multimodal fusion, optimized by a Deep Momentum Gradient Descent (DeepMomentum).

promote the learning of lazy ones. However, such heuristic approaches can rarely achieve extensive balance, which even impairs the learning of strong modalities. To tackle this issue, several other works (*e.g.*, DGL (Wei et al., 2025a)) seek to theoretically alleviate optimization conflicts, aiming to exploit modality-specific information for more comprehensive multimodal learning.

Although the aforementioned approaches can promote balanced or extensive multimodal learning to a certain extent, their conventional monolithic frameworks fundamentally limit the exploitation of unimodal contribution and performance, as illustrated in Figure 1 (a). Specifically, the frameworks typically rely on coupled gradient-based optimization and integrate discrepant information at a single temporal scale. Under this design, interference and incoordination across modalities are inevitable during training, which in turn degrades the multimodal learning performance.

To address these issues, based on the emerging Nested Learning (NL) paradigm (Behrouz et al., 2025), this paper proposes a novel **M**ultim**o**dal **Ne**s**t**ed Learning framework (MoNet) to nest multimodal joint optimization into several independent sub-processes, enabling decoupled optimization and coordinated fusion, as illustrated in Figure 2. Our MoNet consists of two main components: a Decoupled Multimodal Stable Memory block (DMSM) and an Adaptive Multimodal Coordinated Fusion block (AMCF). More specifically, DMSM constructs an outermost nested level to slowly integrate task-related long-term memories. To be specific, DMSM separates the multimodal encoders and joint classifier as isolated associative memories of NL, which compress task semantics into their model parameters with respective objectives. Benefiting from this, DMSM

can decouple interfering optimization dynamics into independent streams, exploiting modality-specific information for classification prediction. In addition, AMCF forms an inner nested level and leverages a Coordinated Continuum Memory System (CCMS) to mediate multimodal fusion. Specifically, CCMS explicitly decomposes the error signal from the outer optimization and allocates it to its multi-timescale memory layers. With this design, the dominant signal from the strong modality is adaptively integrated into the slow layer, while lazy yet informative modalities are prioritized to accelerate processing with fast layers, embracing balanced multimodal fusion. Benefiting from the decoupled optimization and balanced fusion enabled by DMSM and AMCF, our MoNet achieves superior joint learning performance across diverse datasets, as shown in Figure 1 (b). The main contributions of this paper are summarized as follows:

- We propose a Multimodal Nested Learning Framework (MoNet), which reformulates an imbalanced multimodal learning framework into nested sub-processes for decoupled and coordinated learning. To the best of our knowledge, this work could be one of the first attempts to address multimodal tasks in a nested learning paradigm.
- A Decoupled Multimodal Stable Memory block (DMSM) is proposed to decouple multimodal optimization to exploit modality-specific information independently.
- An Adaptive Multimodal Coordinated Fusion block (AMCF) is presented to coordinate multimodal information integration across multi-timescale memory layers for balanced fusion.
- We conduct extensive comparison experiments on eight datasets across three tasks. Our MoNet shows its superiority without bells and whistles.

## 2. Related Work

### 2.1. Imbalanced Multimodal Learning

Multimodal learning seeks to jointly exploit multiple modalities (Feng et al., 2026b; Su et al., 2026; Li et al., 2025c; Su et al., 2025), media (Pu et al., 2025b; Li et al., 2025b; Liu et al., 2025), and views (Li et al., 2025a; He et al., 2024; 2026), such as audio-visual (Tian et al., 2018), vision-language (Chen et al., 2026; Liu et al., 2026; Wang et al., 2025), visible-infrared (Li et al., 2026), 2D-3D (Yang et al., 2025; Feng et al., 2025) data, to enhance real-world perception beyond unimodal approaches. However, several studies (Winterbottom et al., 2020) have observed that incorporating additional modalities does not always lead to desirable performance gains. Recent works (Huang et al., 2022) have shown empirically and theoretically that a strong modality governs the optimization process, causing the model to neglect other lazy modalities and resulting in imbalanced multimodal learning. To mitigate this issue, early methods (Li et al., 2023; Xu et al., 2023) directly attempt to suppress the optimization gradients of the strong modality, increasing the contribution of lazy modalities for more balanced multimodal learning. In addition, several methods (Duan et al., 2025) calibrate the optimization of different modalities to alleviate inherent conflicts in the optimization processes, further facilitating comprehensive learning across modalities. In this work, we reorganize model components into multiple non-interfering nested levels for decoupled and coordinated multimodal learning.

### 2.2. Nested Learning

Motivated by neuroscientific evidence of multi-timescale learning in the human brain (Frey & Morris, 1997; Dudai et al., 2015; Pu et al., 2025c), Nested Learning (NL) (Behrouz et al., 2025) is proposed to address limitations of deep learning in continual adaptation and memory consolidation. NL models a learning system as a hierarchy of nested associative memory modules, where each level compresses its own context flow under a distinct objective and operates at a different update frequency. In contrast to conventional associative memory models and energy-based learning frameworks, this formulation generalizes and unifies prior paradigms such as meta-learning, learned optimizers (Behrouz et al., 2025), and in-context learning (Brown et al., 2020; Feng et al., 2026a), which can be viewed as special cases of two-level nested optimization. Building on this view, the Continuum Memory System (CMS) introduces fast and slow memory components embedded in a nested structure to enhance the diversity and robustness of knowledge retention. From the NL perspective, this paper aims to decouple monolithic multimodal learning frameworks (Peng et al., 2022; Pu et al., 2025a), enabling more effective memory and fusion of heterogeneous multimodal features.

## 3. Method

### 3.1. Preliminaries and Problem Statement

**Nested Learning.** NL models the learning process as a hierarchical system composed of multiple associative memories operating at different temporal scales. Each associative memory $\mathcal{M}$ learns a mapping that compresses the *context* key $\mathcal{K}$ into value $\mathcal{V}$, which can be written as:

$$\mathcal{M}^* = \arg \min_{\mathcal{M}} \mathcal{L}(\mathcal{M}(\mathcal{K}); \mathcal{V}), \tag{1}$$

where $\mathcal{M}^*$ is the optimized associative memory, *objective* $\mathcal{L}(\cdot; \cdot)$ measures the quality of the mapping. NL constructs various nested levels with respective *update frequencies*. Specifically, inner levels perform rapid adaptive learning at high frequencies, while outer levels integrate long-term knowledge at low frequencies, without mutual interference. Through this hierarchical nested structure, the overall learning process is naturally decoupled into multiple relatively independent yet cooperative optimization sub-processes.

**Multimodal Learning Task.** Given a multimodal dataset with $C$ classes, it is denoted as $\mathcal{D} = \{\{\mathcal{X}^j\}_{j=1}^M, \mathcal{Y}\} = \{\{x_i^j\}_{j=1}^M, y_i\}_{i=1}^N$, where $x_i^j \in \mathcal{X}^j$ is $i$-th input data of the $j$-th modality, $y_i \in \mathcal{Y} = \{1, 2, \ldots, C\}$ is $i$-th ground truth label, and $N$ and $M$ are the number of samples and modalities, respectively. This task aims to fuse information from multiple modalities to produce complementary and accurate classification predictions. Specifically, the raw data are first fed into a multimodal encoder $\mathcal{M}_e^j$ to obtain modality-specific features $\{\mathcal{Z}^j\} = \{\{z_i^j\}_{j=1}^M\}_{i=1}^N$. These features are then jointly fused (*i.e.*, concatenation) and passed into a fusion module $\mathcal{M}_f$ to generate fused features $\{\mathcal{Z}\} = \{z_i\}_{i=1}^N$, which are subsequently input into a classifier $\mathcal{M}_c$ to obtain the final predictions $\hat{\mathcal{Y}} = \{\psi(W_c z_i)\}_{i=1}^N = \{\hat{y}_i\}_{i=1}^N$, where $\psi$ is the *Softmax* function and $W_c$ is the parameter matrix of $\mathcal{M}_c$. However, modality heterogeneity inevitably leads to imbalanced optimization, where strong modalities disproportionately influence learning, while lazy modalities are suppressed. It is commonly referred to as imbalanced multimodal learning, which significantly degrades the performance of joint learning.

To address this issue, we introduce NL and propose the Multimodal Nested Learning Framework (MoNet), as shown in Figure 2, which decouples multimodal joint optimization into several nested sub-processes with independent modality-wise learning at the outer level and coordinated fusion at inner levels. In this framework, we introduce a Decoupled Multimodal Stable Memory block (DMSM) to independently exploit modality-specific task semantics, and an Adaptive Multimodal Coordinated Fusion block (AMCF) to adaptively coordinate multimodal features for fusion.

## 3.2. Decoupled Multimodal Stable Memory

Our DMSM corresponds to the outermost level with the slowest *update frequency* $f_s$ in the NL framework. It is designed to compress multimodal input data into task-related long-term memory, *i.e.*, the stable network parameters. Specifically, DMSM contains two types of associative memories: Multimodal Encoders and a Joint Classifier.

**Multimodal Encoders.** They form a set of modality-specific associative memories. Each encoder operates on the raw data from its corresponding modality as its input *context*, and aims to compress task semantics into the backbone parameters for discriminative feature encoding. To measure the quality of the associative mapping, we independently compute Cross-Entropy (CE) between the unimodal predictions $\hat{\mathcal{Y}}$ and the ground-truth labels $\mathcal{Y}$ as the learning *objective*. Accordingly, taking the $j$-th modality as an example, this associative memory $\mathcal{M}_e^j$ can be formulated as:

$$
\begin{aligned}
\mathcal{M}_e^{j*} &= \arg\min_{\mathcal{M}_e^j} \lambda \mathcal{L}_{ce}(\hat{\mathcal{Y}}^j; \mathcal{Y}), \\
\hat{\mathcal{Y}}^j &= \bar{\mathcal{M}}_c\big(\bar{\mathcal{M}}_f\big(\sigma\big(\{\mathcal{M}_e^j(\mathcal{X}^j)\}_{j=1}^M\big) \odot \boldsymbol{m}^j\big)\big),
\end{aligned}
\tag{2}
$$

where $\mathcal{L}_{ce}$ represents the CE loss, $\lambda$ serves as a weighting trade-off between the encoders and the classifier during the outer-level optimization, $\bar{\mathcal{M}}$ denotes the frozen associative memory that is not affected by the process, $\sigma(\cdot)$ denotes feature concatenation across modalities, $\boldsymbol{m}^j$ is the modality mask vector in which all modality features except the $j$-th one are masked to zero vectors, and $\odot$ denotes element-wise multiplication. Notably, the quality of the associative mapping is not evaluated directly on the output modality-specific features, but instead through their respective contribution to the task objective. This design explicitly encourages the encoders to learn task-related features, enabling them to provide complementary modal information.

**Joint Classifier.** This associative memory takes the fused multimodal features as its *context*, and aims to compress the class-level semantic information contained therein into the classifier parameters for classification prediction. Specifically, we compute CE between the multimodal fusion predictions and the ground-truth labels as the *objective*. Accordingly, this associative memory $\mathcal{M}_c$ is formulated as:

$$
\mathcal{M}_c^* = \arg\min_{\mathcal{M}_c} \mathcal{L}_{ce}(\mathcal{M}_c(\mathcal{Z}); \mathcal{Y}).
\tag{3}
$$

In DMSM, the two aforementioned disjoint associative memories are each optimized via gradient backpropagation using independent momentum-based SGD optimizers. This design decouples the conventional entangled training objective and optimization gradient, effectively eliminating interference between modules and modalities during the encoding and prediction stages.

## 3.3. Adaptive Multimodal Coordinated Fusion

Although DMSM ensures non-interfering multimodal learning, its fixed and slow memory integration is inherently limited in adapting to the competitive multimodal fusion process. To address this limitation, we introduce AMCF for adaptive and coordinated feature fusion. Specifically, AMCF is nested within DMSM and operates conditioned on the multimodal features it produces, seeking to integrate and align them with the outer learning objective.

**Continuum Memory System.** To instantiate AMCF under the NL paradigm, we introduce a Continuum Memory System (CMS) (Behrouz et al., 2025) as a multi-timescale associative memory $\mathcal{M}_f$, which is formally implemented as a cascade of three layers of MLPs operating at different *update frequencies*. At each layer, CMS takes fused features from the previous level as input *context* and learns a task-related semantic correction, which is then applied in a residual manner to update the fused features. Taking the $l$-th layer of CMS (*i.e.*, $\mathcal{S}^{(l)}$) as an example, this process can be formulated as:

$$
\boldsymbol{z}_i^{(l)} = \boldsymbol{z}_i^{(l-1)} + \mathcal{S}^{(l)}(\boldsymbol{z}_i^{(l-1)}),
\tag{4}
$$

where $\boldsymbol{z}_i^{(l)}$ denotes the fused feature of the $i$-th sample after the $l$-th CMS layer. In particular, the feature before the first layer is directly obtained by concatenating the multimodal features produced by the outer nested level, *i.e.*, $\boldsymbol{z}_i^{(0)} = \sigma(\{\boldsymbol{z}_i^j\}_{j=1}^M)$. For CMS optimization, it receives the task-semantic *error signal* projected from the outer to the inner nested level, *i.e.*, $\boldsymbol{\delta}$, which is formulated as:

$$
\boldsymbol{\delta} = \frac{1}{|\mathcal{C}^l|}\sum_{i\in\mathcal{C}^l} W_c^\top(\nabla_{\boldsymbol{o}_i}\mathcal{L}_{ce}) = \frac{1}{|\mathcal{C}^l|}\sum_{i\in\mathcal{C}^l} W_c^\top(\hat{\boldsymbol{y}}_i - \boldsymbol{y}_i),
\tag{5}
$$

where $\boldsymbol{o}_i = W_c \boldsymbol{z}_i$, $\mathcal{C}^l$ denotes a chunk of the $l$-th CMS layer that accumulates samples under its specific *update frequency*, which can be understood as batches with different update rates and sizes. Detailed explanation and derivations of this process could be found in our Appendix.

Each CMS layer computes $L_2$ regression loss between the output fused features and task-semantic corrected features. It measures the quality of this associative mapping, serving as the *objective* to teach the optimization, written as:

$$
\mathcal{L}_\delta' = \frac{1}{|\mathcal{C}^l|}\sum_{l=1}^L \sum_{i\in\mathcal{C}^l}\left\|\underbrace{\big(\boldsymbol{z}_i^{(l-1)} + \mathcal{S}^{(l)}(\boldsymbol{z}_i^{(l-1)})\big)}_{\text{CMS Fused Feature}} - \underbrace{\big(\boldsymbol{z}_i^{(l-1)} - \boldsymbol{\delta}\big)}_{\text{Corrected Feature}}\right\|_2^2,
$$

where $L = 3$ denotes the number of CMS layers, $\|\cdot\|_2^2$ means squared error.

**Coordinated Continuum Memory System.** However, updating CMS with the single and undifferentiated *error signal*

is insufficient to mitigate imbalanced modality contributions during fusion. To this end, we propose a novel Coordinated Continuum Memory System (CCMS), an extension of CMS, which explicitly decomposes the *error signal* and distributes it across layers used in CMS for coordinated fusion refinement. Specifically, following Exponential Moving Average (EMA) filtering, CCMS maintains a short-term error memory $\boldsymbol{\delta}_s$ with decay factor $\beta_1$ and a long-term error memory $\boldsymbol{\delta}_l$ with $\beta_2$, which can be defined as:

$$\boldsymbol{\delta}_s = \beta_1 \bar{\boldsymbol{\delta}}_s + (1 - \beta_1)\boldsymbol{\delta}, \quad \boldsymbol{\delta}_l = \beta_2 \bar{\boldsymbol{\delta}}_l + (1 - \beta_2)\boldsymbol{\delta}, \quad (6)$$

where $\bar{\boldsymbol{\delta}}_s$ and $\bar{\boldsymbol{\delta}}_l$ denote the historical error memory prior to the arrival of $\boldsymbol{\delta}$. Based on the statistics, CCMS decomposes the *error signal* into three complementary components:

**1)** Swift signal $\boldsymbol{\delta}^{(1)}$ captures rapidly varying correction trends in the *error signal*, formulated as $\boldsymbol{\delta}^{(1)} = \boldsymbol{\delta} - \boldsymbol{\delta}_s$. Specifically, it isolates the instantaneous deviation of the current error from its short-term average, which is typically induced by strong modalities that provide immediate effective supervision during optimization. Therefore, we attempt to route it to the first slow-update layer at *update frequency* $f_s$, where it is moderated through long-term memory integration to slow down dominant modality signals, thereby alleviating imbalance.

**2)** Consensus signal $\boldsymbol{\delta}^{(2)}$, defined as $\boldsymbol{\delta}^{(2)} = \boldsymbol{\delta}_s - \boldsymbol{\delta}_l$, reflects correction components that persist over intermediate temporal scales. It isolates shared cues that are consistently aligned across modalities, encoding consensus corrections arising from multimodal fusion. To effectively capture such consensus, CCMS distributes this signal to a fast-update layer at update frequency $f_f = \gamma f_s$, enabling rapid consensus formation and coherent multimodal integration.

**3)** Steady signal $\boldsymbol{\delta}^{(3)} = \boldsymbol{\delta}_l$ captures slowly accumulating discrepancies. It captures the long-term trend of the *error signal* via slow exponential averaging, reflecting modalities that convey richer and underleveraged semantics that are prone to being suppressed by simple strong modalities. To exploit the persistent yet informative signal, CCMS assigns it to the last fast-update layer at $f_f$, accelerating memory integration for comprehensive learning.

By decomposing and allocating the *error signal* across CCMS layers, the layer-specific diverse *objective* can be written as:

$$\mathcal{L}_\delta = \frac{1}{|\mathcal{C}^l|} \sum_{l=1}^{L} \sum_{i \in \mathcal{C}^l} \left\| \mathcal{S}^{(l)}\left(\boldsymbol{z}_i^{(l-1)}\right) - \left(-\boldsymbol{\delta}^{(l)}\right) \right\|_2^2. \quad (7)$$

Benefiting from this design, CCMS enables coordinated multimodal fusion, which can be modeled as:

$$\mathcal{M}_f^* = \arg\min_{\mathcal{M}_f} \mathcal{L}_\delta\left(\mathcal{M}_f(\sigma(\{\mathcal{Z}^j\}_{j=1}^{M})); \mathcal{Z}\right). \quad (8)$$

## 3.4. Nested Optimization

Under the NL paradigm, both outer (*i.e.*, DMSM) and inner levels (*i.e.*, AMCF) are optimized by dedicated optimizers, which themselves could be viewed as deeper nested levels. On one hand, the associative memories in DMSM are updated using backpropagation with momentum-based SGD, which can be viewed as a two-level nested structure. Specifically, the outer level maps model parameters to their corresponding update gradients, and the inner level compresses instantaneous gradients into a momentum state. On the other hand, CCMS in AMCF is optimized through the Deep Momentum Gradient Descent (Behrouz et al., 2025), which can be interpreted as a deeper nested associative memory on gradients. Similarly, the inner level memorizes the deviation of instantaneous gradients through a momentum state, which is subsequently used to modulate the effective update direction applied to the parameters, thereby driving parameter evolution. Due to space limitations, further discussion of nested optimization is deferred to our Appendix.

## 4. Experiment

To evaluate our MoNet, we conduct extensive comparison experiments across audio-visual, image-text, and 2D-3D recognition tasks on eight multimodal datasets.

### 4.1. Experimental Settings

We compare our MoNet with two vanilla fusion baseline (*i.e.*, Concatenation and Summation) and 8 state-of-the-art methods (*i.e.*, OGM-GE (Peng et al., 2022), AGM (Li et al., 2023), MMCosine (Xu et al., 2023), MMPareto (Wei & Hu, 2024), MLA (Zhang et al., 2024), DGL (Wei et al., 2025a), ARL (Wei et al., 2025b), and InfoReg (Huang et al., 2025)).

For a fair comparison, all models are trained and evaluated on NVIDIA GeForce RTX 3090 GPUs. To mitigate the randomness in experimental results, we report the mean accuracy and standard deviation of both unimodal and multimodal recognition over three random runs. All baseline methods are carefully tuned to their optimal configurations before being evaluated across the three tasks. Specifically, following (Qin et al., 2026), for each task, we adopt the commonly used benchmark datasets and processing protocols. **1)** *Audio-visual recognition* involves the CREMA-D (Cao et al., 2014), AVE (Tian et al., 2018), AVSBench, and VG-GSound50 (Chen et al., 2020) datasets. For audio modality, ResNet18 is adopted for feature extraction. For video modality, three frames are randomly sampled from each video and fed into a ResNet18 to extract visual features. **2)** *Image-text recognition* is conducted on the MVSA (Niu et al., 2016) and FOOD101 (Bossard et al., 2014) datasets. For the images, a ResNet18-based backbone is employed to encode visual representations, while the texts are processed using

*Table 1.* Performance comparison in terms of test accuracy (%) on four audio-visual datasets for audio-visual recognition. Results are reported as mean $\pm$ std over three random runs. The highest fusion recognition performances are shown in **bold**.

| Method | Mode | CREMA-D | AVE | AVSBench | VGGSound50 |
|---|---|---|---|---|---|
| Summation | FUSION | $61.07 \pm 1.30$ | $66.33 \pm 0.85$ | $87.43 \pm 0.90$ | $53.73 \pm 0.48$ |
|  | AUDIO | $53.15 \pm 2.61$ | $53.73 \pm 1.42$ | $76.67 \pm 0.80$ | $40.55 \pm 0.26$ |
|  | VIDEO | $22.18 \pm 2.39$ | $19.32 \pm 1.73$ | $25.36 \pm 0.39$ | $19.04 \pm 0.82$ |
| Concatenation | FUSION | $63.26 \pm 0.91$ | $66.42 \pm 0.54$ | $87.34 \pm 0.45$ | $53.82 \pm 0.63$ |
|  | AUDIO | $54.75 \pm 0.39$ | $53.48 \pm 1.81$ | $77.66 \pm 1.62$ | $40.78 \pm 0.27$ |
|  | VIDEO | $24.51 \pm 1.47$ | $18.33 \pm 0.59$ | $22.89 \pm 2.33$ | $19.33 \pm 0.35$ |
| OGM-GE (Peng et al., 2022) | FUSION | $76.43 \pm 0.84$ | $65.67 \pm 1.02$ | $87.50 \pm 0.67$ | $55.96 \pm 0.21$ |
|  | AUDIO | $57.26 \pm 0.52$ | $50.66 \pm 1.19$ | $74.41 \pm 2.02$ | $38.64 \pm 0.14$ |
|  | VIDEO | $64.92 \pm 1.12$ | $19.82 \pm 1.73$ | $26.37 \pm 2.52$ | $22.21 \pm 0.18$ |
| AGM (Li et al., 2023) | FUSION | $65.68 \pm 0.73$ | $68.32 \pm 0.94$ | $87.93 \pm 0.88$ | $53.94 \pm 0.51$ |
|  | AUDIO | $57.66 \pm 0.48$ | $54.23 \pm 2.54$ | $79.28 \pm 0.90$ | $42.99 \pm 0.23$ |
|  | VIDEO | $26.84 \pm 0.85$ | $17.49 \pm 1.75$ | $24.68 \pm 0.92$ | $20.69 \pm 0.64$ |
| MMCosine (Xu et al., 2023) | FUSION | $63.73 \pm 0.42$ | $67.49 \pm 0.47$ | $89.19 \pm 0.69$ | $54.15 \pm 0.33$ |
|  | AUDIO | $55.29 \pm 1.58$ | $55.72 \pm 0.20$ | $81.89 \pm 0.80$ | $43.50 \pm 0.49$ |
|  | VIDEO | $31.99 \pm 2.50$ | $20.98 \pm 0.31$ | $35.59 \pm 3.89$ | $25.65 \pm 2.71$ |
| MMPareto (Wei & Hu, 2024) | FUSION | $69.53 \pm 1.20$ | $69.73 \pm 0.62$ | $91.44 \pm 0.93$ | $59.06 \pm 0.13$ |
|  | AUDIO | $61.02 \pm 1.43$ | $61.36 \pm 1.12$ | $81.80 \pm 0.95$ | $45.02 \pm 0.09$ |
|  | VIDEO | $46.64 \pm 2.01$ | $33.17 \pm 1.12$ | $54.05 \pm 0.40$ | $36.05 \pm 0.69$ |
| MLA (Zhang et al., 2024) | FUSION | $74.42 \pm 0.89$ | $68.74 \pm 0.47$ | $84.59 \pm 1.34$ | $58.21 \pm 0.77$ |
|  | AUDIO | $59.14 \pm 0.88$ | $64.10 \pm 0.47$ | $75.81 \pm 2.65$ | $46.91 \pm 0.46$ |
|  | VIDEO | $65.05 \pm 0.29$ | $32.18 \pm 1.24$ | $40.99 \pm 0.17$ | $35.16 \pm 0.92$ |
| DGL (Wei et al., 2025a) | FUSION | $79.40 \pm 1.59$ | $68.82 \pm 2.34$ | $91.98 \pm 0.33$ | $57.40 \pm 0.22$ |
|  | AUDIO | $61.88 \pm 2.26$ | $63.60 \pm 1.52$ | $83.97 \pm 1.64$ | $48.39 \pm 0.57$ |
|  | VIDEO | $72.96 \pm 2.34$ | $24.87 \pm 1.33$ | $63.54 \pm 0.77$ | $39.44 \pm 0.47$ |
| ARL (Wei et al., 2025b) | FUSION | $78.87 \pm 0.49$ | $69.87 \pm 0.44$ | $91.94 \pm 0.17$ | $60.19 \pm 0.79$ |
|  | AUDIO | $60.24 \pm 1.32$ | $62.50 \pm 1.61$ | $83.29 \pm 0.29$ | $46.31 \pm 0.45$ |
|  | VIDEO | $69.19 \pm 1.49$ | $37.50 \pm 1.39$ | $56.23 \pm 1.12$ | $38.91 \pm 0.15$ |
| InfoReg (Huang et al., 2025) | FUSION | $74.77 \pm 0.77$ | $69.89 \pm 0.19$ | $91.35 \pm 0.51$ | $59.46 \pm 0.32$ |
|  | AUDIO | $59.13 \pm 2.72$ | $63.25 \pm 0.97$ | $81.80 \pm 0.51$ | $46.42 \pm 0.36$ |
|  | VIDEO | $61.00 \pm 0.62$ | $31.44 \pm 0.93$ | $52.88 \pm 0.55$ | $35.43 \pm 0.72$ |
| **Our Method** | FUSION | $\mathbf{83.11} \pm 0.46$ | $\mathbf{71.23} \pm 2.17$ | $\mathbf{93.06} \pm 0.42$ | $\mathbf{60.90} \pm 0.23$ |
|  | AUDIO | $59.86 \pm 1.12$ | $63.10 \pm 0.85$ | $80.50 \pm 0.34$ | $46.14 \pm 1.41$ |
|  | VIDEO | $74.33 \pm 1.15$ | $39.39 \pm 1.19$ | $65.50 \pm 1.06$ | $40.88 \pm 1.18$ |

pretrained GloVe embeddings followed by a Bi-GRU. **3)** *2D-3D recognition* includes the 3D MNIST and ModelNet datasets. For 2D images, a CNN is used for 3D MNIST, whereas a multi-channel ResNet18 is used for ModelNet. For 3D point clouds, DGCNN is consistently used for both datasets. Due to space limitations, the introduction of the datasets and evaluation metrics, implementation details of MoNet, and analysis of $\beta_1$ and $\beta_2$ are provided in our Appendix.

## 4.2. Comparison with the State-of-the-Arts

We apply multimodal recognition on the eight datasets to evaluate our MoNet and the baselines. The audio-visual recognition experimental results are reported in Table 1, image-text and 2D-3D recognition results are reported in Table 2. From these experimental results, we could obtain the following observations: **1)** Optimization imbalance across modalities is pervasive in multimodal learning tasks, leading to severe performance degradation. Vanilla joint learning methods struggle to balance optimization across modali-

ties, resulting in underfitting of the lazy modality and a significant decline in fusion recognition performance. **2)** Existing heuristic optimization modulation methods (*e.g.*, OGM-GE, InfoReg, *etc.*) fail to deliver consistently strong performance across datasets and tasks. In addition, other conflict-alleviation methods (*e.g.*, DGL, *etc.*), while capable of achieving high unimodal performance, are constrained by inherently coupled optimization frameworks, causing the fusion performance to be inevitably impacted. **3)** Our MoNet achieves superior fusion recognition performance while simultaneously delivering enhanced unimodal performance. This demonstrates the intrinsic capability of NL in enabling decoupled yet coordinated learning across modalities.

## 4.3. Ablation and Parameter Study

We first conduct an ablation study to evaluate the contribution of each proposed component to our MoNet, as shown in Table 3. Specifically, we first flatten the overall nested framework of MoNet into a conventional joint optimization one and fix the update frequency to 1. We then ablate

*Table 2.* Performance comparison on MVSA and FOOD101 for image-text recognition, and on 3D MNIST and ModelNet for 2D-3D recognition. Results are reported as mean ± std over three random runs. The highest fusion recognition performances are shown in **bold**.

| Method | Mode | MVSA | FOOD101 | 3D MNIST | ModelNet |
|---|---|---|---|---|---|
| Summation | FUSION | $72.19 \pm 0.92$ | $87.58 \pm 0.04$ | $96.80 \pm 0.50$ | $91.33 \pm 0.25$ |
| | IMAGE/2D | $28.45 \pm 0.36$ | $29.42 \pm 0.16$ | $32.00 \pm 10.1$ | $87.20 \pm 0.26$ |
| | TEXT/3D | $65.64 \pm 0.55$ | $82.40 \pm 0.09$ | $96.70 \pm 0.60$ | $55.63 \pm 1.81$ |
| Concatenation | FUSION | $71.16 \pm 0.55$ | $87.50 \pm 0.10$ | $96.80 \pm 0.30$ | $91.32 \pm 0.35$ |
| | IMAGE/2D | $29.67 \pm 2.16$ | $30.11 \pm 0.38$ | $50.19 \pm 0.41$ | $87.12 \pm 0.26$ |
| | TEXT/3D | $61.72 \pm 0.55$ | $87.50 \pm 0.10$ | $87.12 \pm 1.47$ | $50.19 \pm 0.73$ |
| OGM-GE (Peng et al., 2022) | FUSION | $70.84 \pm 1.05$ | $87.49 \pm 0.07$ | $97.30 \pm 0.22$ | $91.10 \pm 0.36$ |
| | IMAGE/2D | $29.61 \pm 0.96$ | $30.39 \pm 0.24$ | $62.53 \pm 8.57$ | $85.67 \pm 0.91$ |
| | TEXT/3D | $60.18 \pm 1.77$ | $81.66 \pm 0.13$ | $96.67 \pm 0.47$ | $53.63 \pm 1.99$ |
| AGM (Li et al., 2023) | FUSION | $67.63 \pm 0.51$ | $87.18 \pm 0.15$ | $97.03 \pm 0.21$ | $90.86 \pm 0.28$ |
| | IMAGE/2D | $30.12 \pm 1.54$ | $30.30 \pm 0.48$ | $80.43 \pm 4.05$ | $80.39 \pm 0.56$ |
| | TEXT/3D | $42.83 \pm 1.03$ | $79.57 \pm 0.16$ | $95.86 \pm 0.15$ | $78.25 \pm 0.22$ |
| MMCosine (Xu et al., 2023) | FUSION | $66.41 \pm 0.51$ | $88.73 \pm 0.12$ | $97.23 \pm 0.12$ | $91.17 \pm 0.68$ |
| | IMAGE/2D | $37.76 \pm 2.32$ | $43.59 \pm 0.16$ | $74.00 \pm 7.73$ | $88.80 \pm 1.04$ |
| | TEXT/3D | $54.66 \pm 1.00$ | $82.89 \pm 0.08$ | $97.07 \pm 0.06$ | $76.50 \pm 4.50$ |
| MMPareto (Wei & Hu, 2024) | FUSION | $72.04 \pm 1.18$ | $89.24 \pm 0.12$ | $98.23 \pm 0.32$ | $91.60 \pm 0.24$ |
| | IMAGE/2D | $50.05 \pm 2.58$ | $49.60 \pm 0.04$ | $97.07 \pm 0.35$ | $89.51 \pm 0.49$ |
| | TEXT/3D | $74.63 \pm 1.10$ | $84.06 \pm 0.05$ | $96.87 \pm 0.68$ | $86.47 \pm 0.54$ |
| MLA (Zhang et al., 2024) | FUSION | $61.87 \pm 1.58$ | $89.30 \pm 0.43$ | $98.16 \pm 0.35$ | $91.72 \pm 1.25$ |
| | IMAGE/2D | $51.57 \pm 1.76$ | $51.70 \pm 2.36$ | $97.50 \pm 0.26$ | $90.27 \pm 0.92$ |
| | TEXT/3D | $68.80 \pm 0.53$ | $83.53 \pm 0.06$ | $96.37 \pm 0.51$ | $88.18 \pm 0.49$ |
| DGL (Wei et al., 2025a) | FUSION | $69.62 \pm 1.37$ | $88.24 \pm 0.01$ | $97.10 \pm 0.37$ | $91.89 \pm 0.40$ |
| | IMAGE/2D | $50.80 \pm 0.78$ | $56.51 \pm 0.22$ | $94.37 \pm 3.94$ | $89.87 \pm 0.30$ |
| | TEXT/3D | $51.32 \pm 0.09$ | $67.51 \pm 0.76$ | $96.57 \pm 0.49$ | $86.86 \pm 0.58$ |
| ARL (Wei et al., 2025b) | FUSION | $70.26 \pm 1.11$ | $88.39 \pm 0.20$ | $97.99 \pm 0.21$ | $86.76 \pm 0.15$ |
| | IMAGE/2D | $51.25 \pm 2.65$ | $56.32 \pm 0.10$ | $97.26 \pm 0.12$ | $91.88 \pm 0.46$ |
| | TEXT/3D | $51.32 \pm 0.09$ | $67.56 \pm 0.84$ | $97.46 \pm 0.18$ | $90.08 \pm 0.25$ |
| InfoReg (Huang et al., 2025) | FUSION | $74.05 \pm 0.79$ | $89.20 \pm 0.15$ | $97.93 \pm 0.70$ | $89.72 \pm 1.11$ |
| | IMAGE/2D | $50.67 \pm 1.10$ | $40.37 \pm 0.24$ | $97.67 \pm 0.40$ | $86.10 \pm 1.46$ |
| | TEXT/3D | $74.57 \pm 1.35$ | $84.26 \pm 0.03$ | $94.43 \pm 2.54$ | $82.32 \pm 1.07$ |
| **Our Method** | FUSION | $\mathbf{74.25} \pm 0.55$ | $\mathbf{90.42} \pm 0.16$ | $\mathbf{98.47} \pm 0.17$ | $\mathbf{92.50} \pm 0.30$ |
| | IMAGE/2D | $50.93 \pm 0.75$ | $54.14 \pm 0.46$ | $97.47 \pm 0.24$ | $91.63 \pm 0.31$ |
| | TEXT/3D | $72.19 \pm 0.40$ | $84.91 \pm 0.09$ | $97.47 \pm 0.17$ | $86.18 \pm 0.20$ |

*Table 3.* Ablation studies for overall, outer, and inner nested modules of our MoNet on CREMA-D. $\Delta \downarrow$ denotes the decrease in fusion recognition accuracy. *w/o* stands for without use.

| | Method | Audio | Video | Fusion | $\Delta \downarrow$ |
|---|---|---|---|---|---|
| | Full | $59.86 \pm 1.12$ | $74.33 \pm 1.15$ | $\mathbf{83.11} \pm 0.46$ | — |
| Overall | *w/o* NL | $54.75 \pm 0.39$ | $24.51 \pm 0.39$ | $63.26 \pm 0.91$ | 19.85 |
| | $f_f = f_s = 1$ | $61.47 \pm 0.96$ | $74.73 \pm 0.99$ | $81.59 \pm 1.16$ | 1.52 |
| Outer | *w/o* DMSM | $27.87 \pm 4.88$ | $16.40 \pm 0.19$ | $31.14 \pm 3.25$ | 51.97 |
| | *w/o* AMCF | $61.88 \pm 2.26$ | $72.96 \pm 2.34$ | $79.40 \pm 1.59$ | 3.71 |
| Inner | *w/o* CCMS | $61.81 \pm 0.11$ | $75.42 \pm 0.51$ | $81.42 \pm 0.31$ | 1.69 |
| | *w/o* EMA | $61.71 \pm 0.71$ | $75.58 \pm 0.23$ | $82.17 \pm 0.06$ | 0.94 |

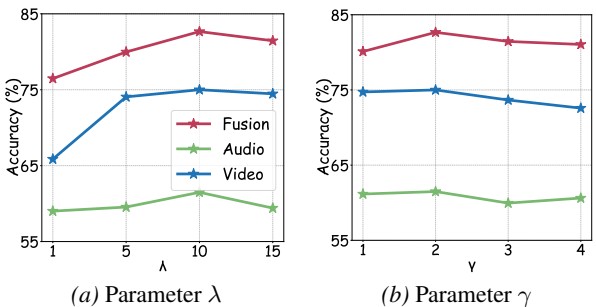

*(a) Parameter $\lambda$*    *(b) Parameter $\gamma$*

*Figure 3.* Fusion, Audio, and Video recognition performance versus different values of $\lambda$ and $\gamma$ of our MoNet on CREMA-D.

the outer-layer DMSM and the inner-layer AMCF by flattening them and removing their independent optimization, respectively. Finally, we ablate the other components (*i.e.*, CCMS and EMA) of AMCF. All structural ablations replace the original designs with generic network layers to avoid confounding effects from parameter volume. From the results, **1)** MoNet without any component will drop fusion recognition performance, which indicates that each component contributes to our method. **2)** After removing the nested structure, MoNet suffers performance degradation.

Especially after removing DMSM, entangled optimization not only impairs outer modality-specific learning but also severely implicates the inner AMCF, leading to even underfitting. These all demonstrate that the NL eliminates cross-modal interference and facilitates coordinated fusion. **3)** Without AMCF (including its CCMS and EMA), unimodal performance may slightly improve, while joint performance degrades. The removal of fusion coordination exposes in-

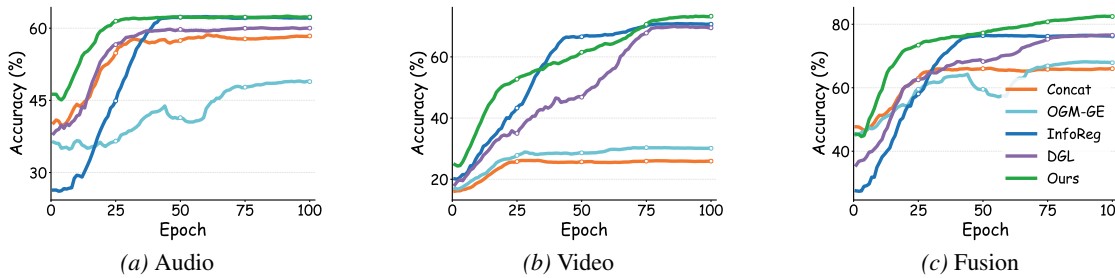

*(a)* Audio      *(b)* Video      *(c)* Fusion

*Figure 4.* Performance comparison across the concatenation method (Concat), OGM-GE, InfoReg, DGL, and our MoNet. (a) - (c) illustrate the audio, video, and fused recognition accuracy on CREMA-D over the training process, respectively.

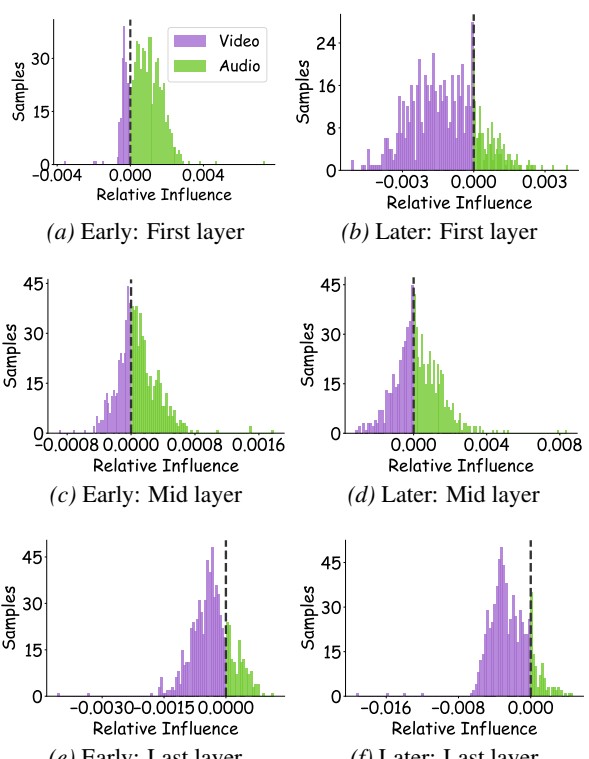

*(a)* Early: First layer      *(b)* Later: First layer

*(c)* Early: Mid layer      *(d)* Later: Mid layer

*(e)* Early: Last layer      *(f)* Later: Last layer

*Figure 5.* Sample distribution of the relative influence of the two modalities on three CCMS layers. (a), (c), and (e) correspond to the early stage of training, while (b), (d), and (f) represent the later stage. Purple (*i.e.*, left of zero) indicates stronger video influence, and green (*i.e.*, right of zero) indicates greater audio influence.

creased yet conflicting semantics within single modalities, which is detrimental to multimodal integration. **4)** Enforcing uniform update frequency prevents MoNet from integrating multi-timescale memory, impairing adaptive fusion.

Moreover, to evaluate the sensitivity of our MoNet to different hyperparameter settings, we plot the recognition performance versus different values of $\lambda$ and $\gamma$ on CREMA-D, as shown in Figure 3. An appropriate $\lambda = 10$ ensures that the encoders and classifier in the outer nested layer each achieve reasonable optimization weights. At the same time, the latter $\gamma = 2$ controls a preferable separation between

high and low update frequencies for our CCMS.

### 4.4. Visualization Analysis

To provide comprehensive insights into our MoNet, we conduct several visualization experiments. Specifically, we present a performance comparison between our MoNet and the DGL, InfoReg, OGM-GE, and concatenation method throughout the training process, as shown in Figure 4. Moreover, to shed light on the rationale behind the adaptive multimodal fusion in CCMS, we visualize the optimization directions of the three layers at different training stages to determine which modality's error signal they align with, as shown in Figure 5. This indicates the *relative influence* of each modality on the corresponding CCMS layers, calculated as $\|\boldsymbol{\delta}^{(l)} - \boldsymbol{\delta}_v\|_2 - \|\boldsymbol{\delta}^{(l)} - \boldsymbol{\delta}_a\|_2$, where $\boldsymbol{\delta}^{(l)}$ is the observed update direction of $l$-th CCMS layer, $\boldsymbol{\delta}_a$ and $\boldsymbol{\delta}_v$ are the audio and visual error signal only calculated with unimodal input based on Equation (5). From the results, we observe that: **1)** Existing baselines fail to achieve comprehensive balance, ultimately leading to inferior multimodal recognition performance. In contrast, our MoNet achieves a more balanced learning process and superior performance, highlighting the intrinsic advantages of isolative optimization and multi-timescale memory integration of NL for multimodal learning. **2)** In the early stage of training, audio modality (*i.e.*, strong modality) is adaptively assigned to the first slower CCMS layer, while the lazy video modality is coordinated to learn faster in the last layer, gradually facilitating optimization balance. At the later stage, the first and last layers dynamically adapt to focus more on the informative and increasingly strong video modality without predefined assignment, and the mid layer continues to incorporate consensus information from both modalities, enabling comprehensive multimodal learning.

## 5. Conclusion

In this paper, we propose a novel Multimodal Nested Learning Framework, namely MoNet, which reformulates multimodal learning under a nested paradigm. Our MoNet integrates two components: the Decoupled Multimodal Sta-

ble Memory block (DMSM) and the Adaptive Multimodal Coordinated Fusion block (AMCF). Specifically, DMSM constructs an outer nested level to disentangle coupled multimodal learning into independent optimization streams, enabling the exploitation of modalities for classification. Moreover, AMCF forms the inner nested level and employs a Coordinated Continuum Memory System (CCMS) to coordinate multimodal integration across multi-timescale memories for balanced fusion. Extensive experiments demonstrate the superiority of our MoNet on eight datasets.

## Acknowledgements

This work was supported in part by the National Key R&D Program of China under Grant 2024YFB4710604; in part by the National Natural Science Foundation of China under Grant 62472295, U25A20534, U25B6003, and 62372315; in part by the Fundamental Research Funds for the Central Universities under Grant CJ202403; in part by Sichuan Science and Technology Planning Project under Grant 2024NS-FTD0049; in part by Central Government's Guide to Local Science and Technology Development Fund under Grant 2025ZYDF101; in part by Chengdu Science and Technology Project under Grants 2025-YF08-00104-GX and 2025-YF05-00169-SN; in part by Foundation Enhancement Program Project (Technology Field Fund) under Grant 2025-JCJQ-JJ-0686; in part by the Luzhou City School-Local-Enterprise-Academy Science and Technology Cooperation Project under Grant 2024XDY200; and in part by Young Science and Technology Scientists Sponsorship Program by CAST - Doctoral Student Special Plan.

## Impact Statement

This paper presents an initial exploration of introducing the emerging nested learning paradigm into multimodal learning tasks. It is a purely technical study that does not involve sensitive data, human subjects, or high-stakes applications, and no specific ethical or societal risks are anticipated beyond those common to the field.

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

# Supplementary Appendix

## Multimodal Nested Learning for Decoupled and Coordinated Optimization

In this appendix, we provide supplementary information and discussion on nested learning concepts, experimental datasets, method details, and additional experiments. More specifically, we first discuss the metrics, formulations, and key concepts involved in our main text. In addition, we further introduce the dataset settings and implementation details of our methods in the experiments, and supplement some additional experimental results and analysis.

## A. Supplementary Introduction and Discussion on Concepts and Formulation

In this section, we first provide additional details on the sources, definitions, and analysis of *Unimodal Contribution* measure and nested learning concepts discussed in Section 1 and Section 3. In addition, we provide detailed derivations of the equations and further theoretical discussions that supplement the main text. We then present a further discussion of the optimizer in Section 3.4 from the perspective of nested learning.

### A.1. Supplementary Definition of Unimodal Contribution

In Section 1 and Figure 1 (a), we follow (Peng et al., 2022), adopting widely used *Unimodal Contribution* to quantify the extent to which each modality supports the learning objective, *i.e.*, classification. A value closer to 1 indicates that the modality better captures task-related semantics. It is worth noting, however, that due to modality heterogeneity and intrinsic data quality, some modalities may not ideally reach a contribution value of 1 even under optimal conditions. For example, the contribution of the $j$-th modality is formulated as (Peng et al., 2022):

$$s^j = \sum_{k=1}^{C} \mathbb{I}_{k=y_i} \left( \mathcal{M}_c \big( \mathcal{M}_f (\sigma(\{\mathcal{M}_e^j(\mathcal{X}^j)\}_{j=1}^M) \odot \boldsymbol{m}^j)) \big) \right)_k, \tag{9}$$

where all the above symbols follow the definitions in the main text. $s^j \in [0, 1]$ is the contribution of $j$-th modality, $\mathbb{I}_{k=y_i}$ is an indicator function that equals 1 if $k = y_i$ and 0 otherwise, $(\cdot)_k$ denotes the predicted probability of the $k$-th class, and $y_i$ denotes the scalar class label corresponding to the one-hot encoded vector $\boldsymbol{y}_i$.

### A.2. Supplementary Formula Derivation

In this section, we provide the complete derivation of Equation (5), which formulates the task-semantic *error signal* propagated from the outer layer to the inner CMS layer.

Consider a $C$-class classification task. Given the fused feature $\boldsymbol{z}_i$ of the $i$-th sample, the classifier produces the logits:

$$\boldsymbol{o}_i = W_c \boldsymbol{z}_i \in \mathbb{R}^C, \tag{10}$$

where $W_c$ denotes the parameter matrix of the classifier. The predicted class probabilities are obtained via the *Softmax* function:

$$\hat{\boldsymbol{y}}_i = \psi(\boldsymbol{o}_i), \qquad \hat{y}_{i,c} = \frac{\exp(o_{i,c})}{\sum_{k=1}^{C} \exp(o_{i,k})}, \tag{11}$$

where $\psi(\cdot)$ denotes the *Softmax* function. The Cross-Entropy (CE) loss for the $i$-th sample is defined as:

$$\mathcal{L}_{ce}^i = -\sum_{c=1}^{C} y_{i,c} \log \hat{y}_{i,c}, \tag{12}$$

where $\boldsymbol{y}_i \in \{0,1\}^C$ is the one-hot encoded ground-truth label. To compute the gradient of the CE loss with respect to the logits $\boldsymbol{o}_i$, we apply the chain rule:

$$\frac{\partial \mathcal{L}_{ce}^i}{\partial o_{i,c}} = \sum_{k=1}^{C} \frac{\partial \mathcal{L}_{ce}^i}{\partial \hat{y}_{i,k}} \frac{\partial \hat{y}_{i,k}}{\partial o_{i,c}}. \tag{13}$$

The partial derivative of the CE loss with respect to $\hat{y}_{i,k}$ is:

$$\frac{\partial \mathcal{L}_{ce}^i}{\partial \hat{y}_{i,k}} = -\frac{y_{i,k}}{\hat{y}_{i,k}}, \tag{14}$$

and the Jacobian of the softmax function is given by:

$$\frac{\partial \hat{y}_{i,k}}{\partial o_{i,c}} = \hat{y}_{i,k}(\delta_{kc} - \hat{y}_{i,c}), \tag{15}$$

where $\delta_{kc}$ denotes the Kronecker delta, written as:

$$\delta_{kc} = \begin{cases} 1, & k = c, \\ 0, & k \neq c. \end{cases} \tag{16}$$

Combining the above terms yields:

$$\frac{\partial \mathcal{L}_{ce}^i}{\partial o_{i,c}} = \hat{y}_{i,c} - y_{i,c}. \tag{17}$$

Therefore, the gradient of the CE loss with respect to the logits can be written in vector form as:

$$\nabla_{\boldsymbol{o}_i} \mathcal{L}_{ce} = \hat{\boldsymbol{y}}_i - \boldsymbol{y}_i. \tag{18}$$

Since the logits are linearly generated from the fused features by $\boldsymbol{o}_i = W_c \boldsymbol{z}_i$, the gradient of the loss with respect to $\boldsymbol{z}_i$ is obtained via the chain rule:

$$\nabla_{\boldsymbol{z}_i} \mathcal{L}_{ce} = W_c^\top \nabla_{\boldsymbol{o}_i} \mathcal{L}_{ce} = W_c^\top (\hat{\boldsymbol{y}}_i - \boldsymbol{y}_i). \tag{19}$$

This term represents the task-semantic *error signal* projected from the outer classification layer to the inner CMS layer.

In CMS optimization, samples are accumulated into a chunk $\mathcal{C}^l$ at the $l$-th CMS layer according to its specific update frequency. The aggregated error signal used to update the CMS parameters is defined as:

$$\boldsymbol{\delta} = \frac{1}{|\mathcal{C}^l|} \sum_{i \in \mathcal{C}^l} W_c^\top (\hat{\boldsymbol{y}}_i - \boldsymbol{y}_i), \tag{20}$$

which exactly corresponds to Equation (5).

## A.3. Theoretical Discussion of CCMS Signal Decomposition

In this section, we provide a concise theoretical justification for the proposed CCMS error signal decomposition. Recall that the task-semantic error signal $\boldsymbol{\delta}$ defined in Equation (5) is computed as the projection of the classification error onto the fused feature space. Under standard stochastic optimization assumptions, $\boldsymbol{\delta}$ is an unbiased estimator of the gradient of the classification loss with respect to the classifier input, *i.e.*,

$$\mathbb{E}[\boldsymbol{\delta}] = W_c^\top \nabla_{\boldsymbol{o}} \mathcal{L}_{ce}, \tag{21}$$

where $\boldsymbol{o} = W_c \boldsymbol{z}$.

CCMS applies Exponential Moving Average (EMA) filtering with decay factors $\beta_1 < \beta_2$ to obtain the short-term and long-term error memories, written as:

$$\boldsymbol{\delta}_s = \beta_1 \bar{\boldsymbol{\delta}}_s + (1 - \beta_1)\boldsymbol{\delta}, \quad \boldsymbol{\delta}_l = \beta_2 \bar{\boldsymbol{\delta}}_l + (1 - \beta_2)\boldsymbol{\delta}. \tag{22}$$

These quantities correspond to low-pass filtered versions of $\boldsymbol{\delta}$ at different temporal scales. Accordingly, the CCMS signal decomposition can be written as

$$\boldsymbol{\delta} = (\boldsymbol{\delta} - \boldsymbol{\delta}_s) + (\boldsymbol{\delta}_s - \boldsymbol{\delta}_l) + \boldsymbol{\delta}_l = \boldsymbol{\delta}^{(1)} + \boldsymbol{\delta}^{(2)} + \boldsymbol{\delta}^{(3)}, \tag{23}$$

which is an exact algebraic identity. Each component isolates a complementary temporal band of the task error, while their summation preserves the complete supervisory signal. Under a local linearization of the classifier output, optimizing the CCMS objective in Equation (8) corresponds to applying temporally factorized correction terms to the fused feature $\boldsymbol{z}$. Assigning $\boldsymbol{\delta}^{(l)}$ to CMS layers with distinct update frequencies therefore realizes a multi-timescale preconditioning of the same gradient signal, improving optimization stability and balancing learning from different modal data.

It is worth noting that this decomposition introduces no additional modality-specific supervision or heuristic weighting. Instead, it leverages the intrinsic temporal statistics of the task error signal, providing a principled and efficient mechanism for coordinated multimodal fusion within the nested learning framework. The analysis assumes that the task error signal $\boldsymbol{\delta}_t$ evolves smoothly over time, which is commonly satisfied in practical multimodal optimization. To be specific, formally, it need to satisfy:

$$\mathbb{E}[\|\boldsymbol{\delta}_{t+1} - \boldsymbol{\delta}_t\|_2] \leq \epsilon, \tag{24}$$

for a small $\epsilon > 0$, where $t$ denotes the optimization step, and $\|\cdot\|_2$ denotes the Euclidean ($L_2$) norm. This condition naturally arises from batch-wise or chunk-wise averaging, the Lipschitz continuity of the classifier, and the slow update dynamics of the outer-level memories. Moreover, $\boldsymbol{\delta}_t$ exhibits temporal correlation,

$$\mathbb{E}[\langle \boldsymbol{\delta}_t, \boldsymbol{\delta}_{t-\tau} \rangle] > 0, \tag{25}$$

for some $\tau > 0$, ensuring that EMA-based filtering captures meaningful task-relevant trends rather than transient stochastic noise.

### A.4. Conceptual Discussion of Nested Learning

In this section, following (Behrouz et al., 2025), we provide supplementary discussions on the specific meanings of key terms in nested learning to facilitate a comprehensive understanding of the paper.

**1) Nested Learning (NL).**

NL reinterprets learning systems as multi-scale computational processes, rather than single optimization events. From this perspective, a model is defined not only by a parameterized function, but also by a hierarchy of update rules operating at different temporal scales. More specifically, NL models learning as nested outer and inner adaptation levels, where slower outer levels generate structured signals (*e.g.*, losses, errors, or teaching signals) that guide the learning dynamics of fast inner levels. Each level optimizes its own objective under a constrained update frequency, while implicitly shaping the learning dynamics of other levels. In this view, architectural depth aligns with optimization depth, *i.e.*, introducing additional nested levels increases algorithmic depth, enabling models to learn how to learn across progressively slower time scales.

**2) Associative memory.**

From the NL viewpoint, every level in the system functions as an associative memory that compresses its incoming context into a parametric or non-parametric state. Associative memory is defined as a mapping from keys to values, learned by minimizing a reconstruction or alignment objective:

$$\mathcal{M}^* = \arg\min_{\mathcal{M}} \mathcal{L}(\mathcal{M}(\mathcal{K}); \mathcal{V}), \tag{26}$$

where where $\mathcal{M}^*$ is the optimized associative memory, $\mathcal{L}(\cdot; \cdot)$ measures the quality of the mapping, $(\mathcal{K}; \mathcal{V})$ are key-value pairs drawn from the level-specific context flow. In NL, both neural layers and optimizers (*e.g.*, momentum, Adam) are associative memories. To be specific, layers associate inputs with representations or prediction errors, while optimizers associate gradients with parameter updates. Crucially, nesting arises because the context of a slower memory (*e.g.*, model parameters) is produced by faster memories (*e.g.*, token-level updates or gradient streams). Learning corresponds to acquiring effective associative memories at each level, while memorization corresponds to storing and retrieving these associations within a given time scale.

**3) Update frequency.**

Update frequency serves as the organizing principle of the NL framework. It is defined as the number of state updates performed by a component per unit time. Specifically, in this paper, we regard the update triggered by a single mini-batch as the fastest update frequency, denoted as $f_f$. For components operating at slower update frequencies, *i.e.*, $f_s = f_f/\lambda$, incoming teaching signals that have not yet reached the designated update frequency are temporarily stored in a frequency-aware chunk with capacity corresponding to $\lambda$ mini-batches (*i.e.*, $\lambda \times bs$ samples), where $bs$ denotes the batch size. Once the chunk is filled, indicating that the prescribed update frequency has been reached, the accumulated teaching signals are consumed to update the corresponding associative memory.

Components with higher update frequency $f_A$ adapt rapidly and encode short-lived information, while lower-frequency components with update frequency $f_B$ evolve slowly and store persistent knowledge. Nested learning orders components by their update frequencies, inducing a partial order $A \succ B$ if $f_A > f_B$. For example, in Transformers, attention states update at effectively infinite frequency within a context window, while MLP weights have zero update frequency at inference time. This frequency-based ordering replaces the traditional static notion of short-term versus long-term memory with a dynamic, algorithmic hierarchy, where learning emerges from interactions across multiple temporal scales.

### 4) Continuum Memory System (CMS).

CMS generalizes discrete memory dichotomies by modeling memory as a spectrum of nested associative memories with continuously varying update frequencies. CMS defines a family of memories indexed by frequency, each obeying an update rule. Within nested learning, CMS enables bidirectional knowledge flow. More specifically, fast memories enable rapid adaptation and exploration, while slow memories consolidate information over long horizons.

Importantly, because all memories are embedded in a single nested system, forgotten information at one frequency can be partially reconstructed via interaction with other levels. This continuum view is essential for continual learning, as it allows the system to balance plasticity and stability without rigid architectural boundaries.

### A.5. Additional Discussion of Nested Optimization

In Section 3.4, we employ different optimization strategies for DMSM and AMCF, namely momentum-based SGD and Deep Momentum Gradient Descent (DeepMomentum). Under the NL paradigm, these optimizers can be interpreted as nested associative memories operating on gradient flows. In this subsection, we provide a more explicit formulation to clarify their nested optimization structures.

### 1) Momentum-based SGD as a two-level nested optimization.

Consider the optimization of DMSM parameters, including the multimodal encoders $\mathcal{M}_e^j$ and the joint classifier $\mathcal{M}_c$, whose parameters are collectively denoted as $\theta \in \mathbb{R}^d$. Given the instantaneous gradient of the task objective,

$$\boldsymbol{g}_t = \nabla_\theta \mathcal{L}_{ce}(\theta_t), \tag{27}$$

where $t$ denotes the optimization iteration index, and $\boldsymbol{g}_t = \nabla_\theta \mathcal{L}(\theta_t)$ is the instantaneous gradient evaluated at the current parameter state $\theta_t$. The momentum update and parameter update are defined as:

$$\theta_{t+1} = \theta_t - \eta \boldsymbol{m}_t, \tag{28}$$

$$\boldsymbol{m}_t = \beta \boldsymbol{m}_{t-1} + (1 - \beta) \boldsymbol{g}_t, \tag{29}$$

where $\boldsymbol{m}_t$ denotes the momentum state at iteration $t$, $\beta \in [0, 1)$ is the momentum coefficient, and $\eta > 0$ is the learning rate.

From the NL perspective, Eq. (28) corresponds to the outer-level optimization, where model parameters $\theta$. evolve at a slower time scale and are updated through a temporally aggregated signal $\boldsymbol{m}_t$. In addition, Eq. (29) corresponds to an inner-level associative memory that incrementally compresses the sequence of instantaneous gradients $\{\boldsymbol{g}_\tau\}_{\tau \leq t}$ into a momentum state $\boldsymbol{m}_t$. This inner level operates at a fast *update frequency* and memorizes the surprise of gradients through temporal accumulation.

Consequently, momentum-based SGD exhibits a two-level nested structure, where a fast inner associative memory shapes the update dynamics of a slower outer parameter system through temporal aggregation.

### 2) Deep Momentum Gradient Descent as a deeper nested optimization.

For AMCF, the parameters of CCMS (*i.e.*, the CMS layers $\{\mathcal{S}^{(l)}\}_{l=1}^L$) are optimized using DeepMomentum, which generalizes the above two-level structure into a deeper nested hierarchy of associative memories.

Let $\varphi$ denote the parameters of CCMS and $\boldsymbol{g}_t = \nabla_\varphi \mathcal{L}_\delta(\varphi_t)$ be the instantaneous gradient of the CCMS objective defined in Eq. (7). Here, $\mathcal{L}_\delta$ is constructed from the task-semantic error signal projected by the outer DMSM and does not involve gradients of outer-level parameters. DeepMomentum maintains a hierarchy of momentum states $\{\boldsymbol{m}_t^{(l)}\}_{l=1}^L$ with different update frequencies, which are updated as:

$$\boldsymbol{m}_t^{(1)} = \alpha_1 \boldsymbol{m}_{t-1}^{(1)} + (1 - \alpha_1)\boldsymbol{g}_t, \tag{30}$$

$$\boldsymbol{m}_t^{(l)} = \alpha_l \boldsymbol{m}_{t-1}^{(l)} + (1 - \alpha_l)\,\phi^{(l)}\Big(\boldsymbol{m}_t^{(l-1)}\Big), \quad l = 2, \ldots, L, \tag{31}$$

where $\boldsymbol{m}_t^{(l)}$ denotes the momentum state of the $l$-th nested level at optimization iteration $t$, $\alpha_l \in [0, 1)$ controls the update frequency of the $l$-th momentum memory, and $\phi^{(l)}(\cdot)$ denotes a nonlinear transformation that aligns lower-level momentum information with the parameter space of the $l$-th CCMS layer. The hierarchy of momentum states mirrors the multi-timescale structure of CCMS layers.

The CCMS parameters are then updated using the outermost momentum state:

$$\varphi_{t+1} = \varphi_t - \eta\,\boldsymbol{m}_t^{(L)}. \tag{32}$$

Under the NL paradigm, each momentum state $\boldsymbol{m}_t^{(l)}$ can be interpreted as an inner-level associative memory that processes and compresses gradient information propagated from the outer optimization objective. More specifically, inner levels operate at faster *update frequencies* and facilitate the rapid integration of informative yet weak gradient components, while outer levels evolve more slowly and encode persistent structures that regulate strong modality signals through gradual and stable learning.

From this perspective, DeepMomentum can be viewed as a deeply nested, state-based optimization process, where inner associative memories progressively reshape outer-level error signals across multiple temporal scales. Such deeper nesting naturally aligns with the multi-timescale design of CCMS in AMCF. Compared with conventional momentum-based SGD, this formulation enables AMCF to exploit higher-order temporal dependencies in gradient flows, offering a principled explanation for its improved adaptability and coordination capability in multimodal fusion.

## B. Supplementary Experiments and Analysis

In this section, we first introduce the eight datasets and the evaluation metrics used in our experiments. To enhance reproducibility, we then provide a transparent algorithm of the implementation details of the proposed MoNet. Finally, we supplement the study with more extensive and scalable parameter and visualization analyses, further facilitating understanding of our MoNet.

### B.1. Datasets

We evaluate our approach on eight multimodal datasets from three different tasks. The corresponding dataset statistics are reported in Table 4, followed by detailed descriptions.

*Table 4.* Overview of the datasets employed in our experiments, including task type, number of classes, and data split configurations.

| Task | Dataset | Classes | Data Split (Train / Val / Test) |
|---|---|---|---|
| Audio-Visual | CREMA-D | 6 | 6,697 / 744 / 744 |
| | AVE | 28 | 3,312 / 402 / 401 |
| | AVSBench | 23 | 3,452 / 740 / 740 |
| | VGGSound50 | 50 | 25,954 / – / 2,499 |
| Image-Text | MVSA | 3 | 1,555 / 519 / 518 |
| | FOOD101 | 101 | 62,971 / 22,715 / 5,000 |
| 2D-3D | 3D MNIST | 10 | 5,000 / – / 1,000 |
| | ModelNet | 40 | 9,840 / – / 3,991 |

- **Audio-visual datasets:**

*CREMA-D* (Cao et al., 2014) is a widely adopted audio-visual benchmark for speech emotion recognition. It contains six emotion categories, including anger, happiness, sadness, neutrality, disgust, and fear. The dataset is composed of 7,442 video clips, each lasting approximately 2–3 seconds, collected from 91 actors uttering short phrases. Following (Peng et al., 2022), we preprocess the raw videos and adopt the same data partition for our experiments.

*AVE* (Tian et al., 2018) is an audio-visual event dataset consisting of 4,143 video clips with an average duration of 10 seconds, collected from YouTube. It covers 28 event categories and is commonly used for audio-visual event localization. In this work, we follow (Qin et al., 2026) to preprocess the raw videos and construct a labeled multimodal dataset for classification, using the same training, validation, and testing splits.

*AVSBench* is an audio-visual benchmark designed for segmentation tasks, comprising both single-source and multi-source subsets. We utilize the single-source subset, which includes over 4,000 video clips from 23 categories, and conduct audio-visual classification following its official data split.

*VGGSound* (Chen et al., 2020) is a large-scale audio-visual dataset featuring more than 300 sound event categories collected in unconstrained environments. Different from the official configuration (Chen et al., 2020), we select a subset of 50 categories, referred to as *VGGSound50*, for our experiments. This subset includes 25,954 clips for training and 2,499 clips for testing.

- **Image-text datasets:**

  *MVSA* (Niu et al., 2016) is a multimodal sentiment analysis dataset containing over 2,000 image-text pairs collected from social media platforms. Each pair is annotated with one of three sentiment labels: positive, neutral, or negative. Following (Zhang et al., 2023), we further refine the neutral samples by considering whether a corresponding non-neutral sample exists. As a result, 1,555 pairs are used for training, 518 for validation, and 519 for testing.

  *FOOD101* (Wang et al., 2015) is a large-scale multimodal food dataset comprising more than 100,000 image-text pairs across 101 food categories. The images are collected via Google Image Search, accompanied by corresponding textual descriptions. We adopt the preprocessing and data split protocol described in (Zhang et al., 2023).

- **2D–3D datasets:**

  *3D MNIST* is a small-scale multimodal dataset that contains 6,000 pairs of handwritten digit images and corresponding 3D point clouds, spanning 10 digit categories from 0 to 9. We follow (Feng et al., 2023) to preprocess the data and use the same experimental split.

  *ModelNet* is a widely used 3D object recognition benchmark consisting of thousands of CAD models from 40 categories. Each 3D object is associated with 180 rendered 2D views. In our experiments, we follow (Feng et al., 2023) to preprocess both the 2D images and 3D shapes and adopt the same data partition.

## B.2. Evaluation Metric

In this section, we mainly supplement the performance metrics used in our experiments (*i.e.*, multimodal recognition accuracy and unimodal recognition accuracy). In addition, due to space limitations, the evaluation metrics adopted in Section 4.4 are only briefly described in our main text. We provide detailed formulations and further discussions in this subsection.

**1) Multimodal recognition accuracy.** In this paper, we adopt multimodal recognition accuracy as the primary evaluation metric, which evaluates the overall effectiveness of multimodal joint learning and fusion. Specifically, it measures the ratio of correctly classified samples to the total number of samples when all modalities are jointly involved in the learning process, which can be formulated as:

$$\text{Accuracy} = \frac{N_c}{N} \times 100\%, \tag{33}$$

where $N$ is the total number of samples, and $N_c$ is the number of correct classification samples.

**2) Unimodal recognition accuracy.** Following (Li et al., 2023; Wei et al., 2025a), unimodal recognition accuracy is employed to assess the extent to which task-related semantics are captured by each individual modality under multimodal joint training. Specifically, it is computed by feeding only a single modality into the trained model, while concatenating zero vectors for the remaining modalities, and measuring the resulting prediction accuracy. This metric facilitates the evaluation of unimodal semantic information exploitation.

**3) Relative influence of modalities to CCMS layer.** In our visualization experiments (*i.e.*, Section 4.4), we adopt the following metric to visualize the relative influence of different modalities on each CCMS layer:

$$\|\boldsymbol{\delta}^{(l)} - \boldsymbol{\delta}_v\|_2 - \|\boldsymbol{\delta}^{(l)} - \boldsymbol{\delta}_a\|_2, \tag{34}$$

where $\boldsymbol{\delta}^{(l)}$ denotes the error signal of the current CCMS layer, $\delta_a$ represents the unimodal error signal obtained by feeding only the audio modality while concatenating zero vectors for the video modality, and $\delta_v$ denotes the unimodal error signal obtained by feeding only the video modality while concatenating zero vectors for the audio modality. Specifically, if the video error signal is more consistent with the CCMS layer error signal, *i.e.*,

$$\|\boldsymbol{\delta}^{(l)} - \boldsymbol{\delta}_v\|_2 < \|\boldsymbol{\delta}^{(l)} - \boldsymbol{\delta}_a\|_2, \tag{35}$$

or equivalently,

$$\|\boldsymbol{\delta}^{(l)} - \boldsymbol{\delta}_v\|_2 - \|\boldsymbol{\delta}^{(l)} - \boldsymbol{\delta}_a\|_2 < 0, \tag{36}$$

this indicates that the optimization of the corresponding layer is more strongly influenced by the video modality. Conversely, if

$$\|\boldsymbol{\delta}^{(l)} - \boldsymbol{\delta}_v\|_2 - \|\boldsymbol{\delta}^{(l)} - \boldsymbol{\delta}_a\|_2 > 0, \tag{37}$$

the optimization of the CCMS layer is more strongly influenced by the audio modality.

### B.3. Implementation Details of MoNet

In this work, our MoNet is implemented in PyTorch. Specifically, the momentum SGD optimizer used by our MoNet has a momentum of 0.9 and a weight decay of $1 \times 10^{-4}$. Without loss of generality, we set the slow *update frequency* to $f_s = 1$, *i.e.*, updating once per batch. $\lambda$ and $\gamma$ are set to 10 and 2, respectively, to ensure appropriate optimization control. Accordingly, the fast *update frequency* is given by $f_f = 2$. Following common practice, $\beta_1$ and $\beta_2$ are fixed to 0.7 and 0.95 to provide reasonable short- and long-term error statistics. Overall, the implementation procedure of MoNet could be referred to in Algorithm 1.

---

**Algorithm 1** Main optimization process of our MoNet

---

**Input:** The training multimodal data $\mathcal{D}$, number of modalities $M$, number of samples $N$, maximal epoch number $N_e$ and learning rate $\eta$.
1: Construct independent nested associative memories, including: multimodal encoders $\{\mathcal{M}_e^j\}_{j=1}^M$, CCMS $\mathcal{M}_f = \{\mathcal{S}^{(l)}\}_{l=1}^3$, and a joint classifier $\mathcal{M}_c$.
2: Maintain a short-term error memory $\boldsymbol{\delta}_s$ and a long-term error memory $\boldsymbol{\delta}_l$.
3: **for** i = 1, 2, $\cdots$, $N_e$ **do**
4:     *// Decoupled Multimodal Stable Memory.*
5:     Extract modality-specific features using multimodal encoders $\mathcal{M}_e^j$, obtaining features $\{\mathcal{Z}^j\} = \{\{z_i^j\}_{j=1}^M\}_{i=1}^N$.
6:     Optimize each multimodal encoder according to Equation (2).
7:     Fuse the modality-specific features using CCMS to obtain the fused features $\{\mathcal{Z}\} = \{z_i\}_{i=1}^N$.
8:     Pass the fused features through the joint classifier to obtain the recognition predictions $\hat{\mathcal{Y}} = \{\hat{y}_i\}_{i=1}^N$.
9:     Optimize the joint classifier according to Equation (3), obtaining outer error signal $\boldsymbol{\delta}$.
10:    *// Adaptive Multimodal Coordinated Fusion.*
11:    Decompose $\boldsymbol{\delta}$ into $\boldsymbol{\delta}^{(1)}$, $\boldsymbol{\delta}^{(2)}$, and $\boldsymbol{\delta}^{(3)}$ based on Equation (6), using $\boldsymbol{\delta}_s$ and $\boldsymbol{\delta}_l$.
12:    Optimize the CCMS according to Equation (8).
13: **end for**
**Output:** Optimized MoNet model parameters.

---

### B.4. Supplementary Parameter Analysis

In this section, we further provide a sensitivity analysis of the EMA parameters $\beta_1$ and $\beta_2$. Specifically, $\beta_1$ and $\beta_2$ control the separation between the short-term error memory $\boldsymbol{\delta}_s$ and the long-term error memory $\boldsymbol{\delta}_l$. We plot the recognition performance under different values of $\beta_1$ and $\beta_2$ on the CREMA-D dataset, as shown in Figure 6. From the experimental results, we observe that:

- Compared with other hyperparameters, $\beta_1$ and $\beta_2$ exhibit smoother sensitivity characteristics and maintain a relatively wide insensitive range.

- When $\beta_1$ is set too small or $\beta_2$ is set too large, the performance degrades. This is because the typical components of the error signal can not be properly distinguished, causing imbalanced signals to be assigned to CMS layers with inappropriate update frequencies, thereby impairing coordinated fusion.

- When $\beta_1$ and $\beta_2$ are set to appropriate intermediate values, EMA achieves the best overall performance.

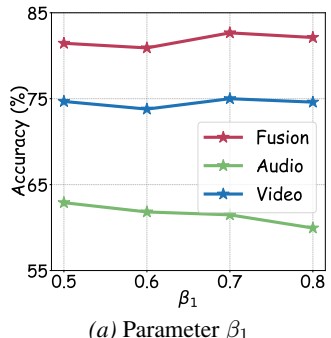 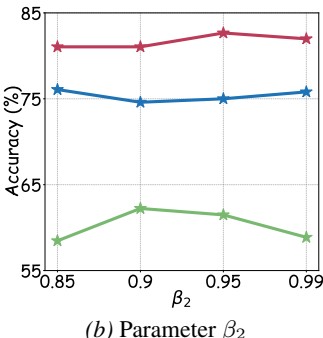

*(a)* Parameter $\beta_1$        *(b)* Parameter $\beta_2$

*Figure 6.* Fusion, Audio, and Video recognition performance versus different values of $\beta_1$ and $\beta_2$ of our MoNet on CREMA-D.

### B.5. Supplementary Visualization Analysis

In this section, we provide more diverse visual analyses to facilitate a deeper understanding of the proposed method. Specifically, we first present performance comparisons between our MoNet and DGL (Wei et al., 2025a), InfoReg (Huang et al., 2025), OGM-GE (Peng et al., 2022), as well as vanilla concatenation fusion throughout the training process on the vision-language MVSA and 2D-3D 3D MNIST datasets. This visualization is shown in Figure 7, which serves as a supplement to Figure 4. Additionally, we compare the unimodal contribution and recognition performance of these methods on MVSA and 3D MNIST. Figure 8 provides additional visualizations that supplement Figure 1 (a). To validate the effectiveness of the proposed DMSM in MoNet, we further visualize the gradient magnitudes received by each modality encoder during training, in comparison with an unmodulated vanilla baseline (*i.e.*, Concat.), as demonstrated in Figure 9. From these results, we derive the following observations:

- Compared with existing methods, MoNet exhibits a more extensive and balanced training process, leading to superior multimodal fusion recognition performance.

- Compared with optimization-modulation and conflict-alleviation approaches, our method encourages more reasonable and balanced unimodal contributions, resulting in improved unimodal recognition performance.

- Compared with the vanilla baseline, MoNet assigns each modality an appropriate optimization rate, enabling more comprehensive unimodal optimization. Specifically, benefiting from the independence of associative memories in nested learning, our DMSM remains robust to imbalanced optimization. Even without explicitly modulating the optimization process, it effectively avoids the dominance of strong modalities over lazy ones, in contrast to the existing vanilla baseline.

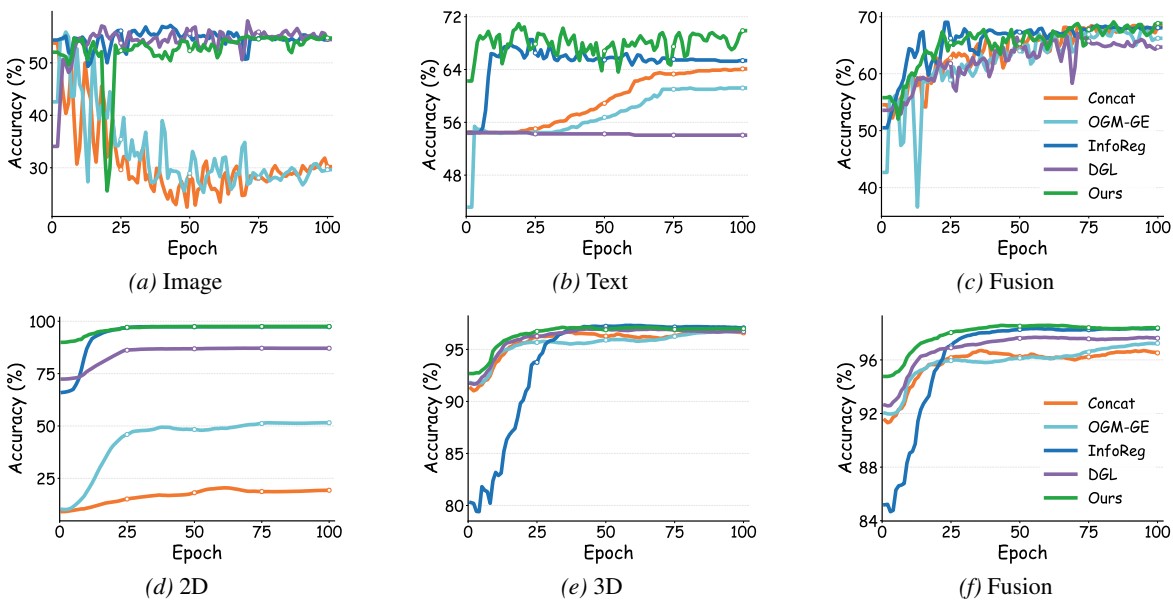

*Figure 7.* Performance comparison across concatenation method (Concat), OGM-GE, InfoReg, DGL, and our MoNet. (a)-(c) illustrate the image, text, and fused recognition accuracy on MVSA validation set over the training process. (d)-(f) respectively illustrate the 2D, 3D, and fused recognition accuracy on 3D MNIST over the entire training process.

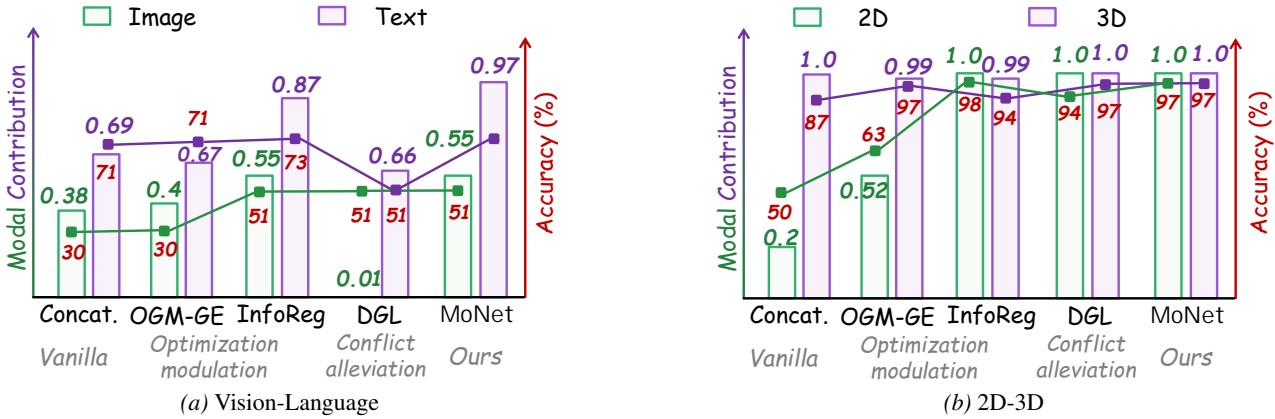

*Figure 8.* Comparison of unimodal contribution (Peng et al., 2022) (*i.e.*, bars) and recognition accuracy (*i.e.*, lines) between our MoNet and DGL, InfoReg, OGM-GE across vision-language and 2D-3D tasks on the MVSA and 3D MNIST datasets.

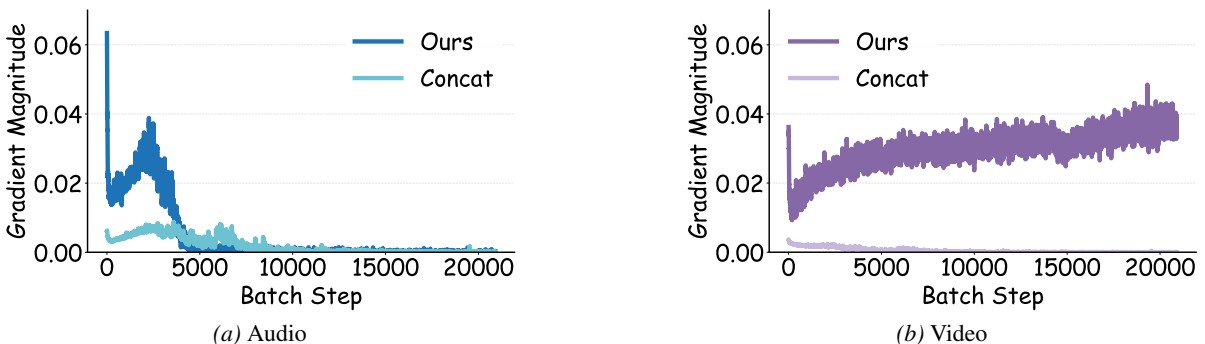

*Figure 9.* Comparison of gradient magnitudes received by the audio and video encoders during training between the vanilla baseline (*i.e.*, Concat.) and our proposed MoNet.

## Limitations

In this paper, we develop a Multimodal Nested Learning Framework (MoNet), which reformulates an imbalanced multimodal learning framework into nested sub-processes. To the best of our knowledge, this work represents one of the first attempts to address multimodal tasks under a nested learning paradigm. As an initial exploration, this paper inevitably has certain limitations in the scope of discussion. In future work, we will continue to explore the broader applicability of nested learning to more downstream multimodal learning tasks.

