# OpenReview forum: "Multimodal Nested Learning for Decoupled and Coordinated Optimization"
_ICML.cc/2026/Conference — ICML 2026 spotlight_

### Official Review · Reviewer_FxR5 · 2026-03-02

**Soundness:** 3
**Presentation:** 3
**Significance:** 4
**Originality:** 3
**Overall Recommendation:** 5
**Confidence:** 5

**Summary:**

The manuscript focuses on imbalanced multimodal learning and proposes a nested learning method, namely Monet, to address this task. Monet comprises two main blocks: DMSM and AMCF. Specifically, DMSM is the outer layer, used to decouple the multimodal classification process, while AMCF is the inner layer, used to coordinate multimodal fusion. This manuscript employs a novel CCMS to achieve adaptive, time-step dynamic feature fusion. Experiments demonstrate the effectiveness of the proposed Monet.

**Compliance With Llm Reviewing Policy:**

Affirmed.

**Final Justification:**

Thanks for the author's response, and my concerns have been well addressed. Thus, I recommend that this paper be accepted.

**Key Questions For Authors:**

If the author could provide further discussion on nested methods and clarification on the experimental settings, method implementation and address my concern, I would be happy to revise my rating.

**Limitations:**

yes

**Strengths And Weaknesses:**

Strengths

This manuscript demonstrates good soundness, originality, and significance in its methodology, and is rich in figures, experiments, and analyses overall. The authors employed DMSM and AMCF to specifically address the problems existing in the current task, achieving SOTA performances in experiments on three multimodal tasks, demonstrating the general effectiveness of the proposed method. However, I still have the following considerations:

 Weaknesses:

1)Unlike previous methods, the description of the innovative framework used by the authors is rather difficult to understand.

2) Furthermore, the decomposition of the three signals in CCMS lacks sufficient theoretical explanation. How can it achieve the effects claimed by the authors?

3) In the ablation experiments, the authors only discussed the case where f is fixed at 1. other cases should be discussed in more detail.

4) The experimental setup and implementation details seem insufficiently detailed. Do the authors follow existing methods, such as [A]?

[A] On-the-fly modulation for balanced multimodal learning.

---

> ### Author Rebuttal · Authors · 2026-03-30
>
> Thanks for your valuable comments. Attached is our point-by-point response.
>
> **Q1: Unlike previous methods, the description of the innovative framework used by the authors is rather difficult to understand.**
>
> **A1:** We thank the reviewer for this valuable feedback. To improve clarity, in the revision we will provide a more comprehensive introduction to the proposed nested learning framework, along with a clearer categorization and review of related work in nested learning (i.e., **Section 2.2**). In addition, in our Method Preliminaries Section (i.e., **Section 3.1**), we will provide a more intuitive explanation of the nested learning framework before introducing the formal definitions, and refine the overall presentation to make it clearer and easier to follow.
>
> **Q2: The decomposition of the three signals in CCMS lacks a sufficient theoretical explanation. How can it achieve the effects claimed by the authors?**
>
> **A2:** We thank the reviewer for this valuable comment. The effect of CCMS comes from decomposing the task-level error signal into multiple time-scale components and optimizing them separately, reshaping the optimization dynamics in a principled way (**see Appendix A.3**).
>
> To be specific, the classification loss is $\mathcal{L}\_{ce}$ and the fused representation is $z$. The task-semantic error $\delta$ used in CCMS satisfies $\mathbb{E}[\delta] = \nabla\_{z} \mathcal{L}\_{ce}$, i.e., it is an unbiased estimator of the gradient signal.
>
> CCMS then constructs short- and long-term error signals via EMA:
> $$
> \delta\_s = \mathrm{EMA}\_{\beta_1}(\delta), \quad \delta\_l = \mathrm{EMA}\_{\beta_2}(\delta),
> $$
>
> and defines fast-/mid-/slow-component:
> $$
> \delta^{(1)} = \delta - \delta_s,\quad \delta^{(2)} = \delta_s - \delta_l,\quad \delta^{(3)} = \delta_l,
> $$
>
> which leads to the exact decomposition:
> $$
> \delta = \delta^{(1)} + \delta^{(2)} + \delta^{(3)} = (\delta - \delta_s) + (\delta_s - \delta_l) + \delta_l.
> $$
>
> This is an algebraic identity that preserves the full supervision signal without approximation.
>
> Under standard assumptions of temporally correlated stochastic gradients, optimizing these components separately is equivalent to a multi-timescale preconditioning of $\nabla_{z} \mathcal{L}_{ce}$, where gradient signals with different temporal characteristics are decoupled and processed independently. By assigning them to nested CMS layers with different update frequencies, CCMS implements this mechanism in practice, leading to more stable optimization and improved modality balancing.
>
> Specifically, CCMS routes $\delta^{(1)}$ to the first slow-update layer, where it is moderated through long-term memory integration to suppress dominant modality signals and alleviate imbalance. To better capture cross-modal consensus, CCMS distributes $\delta^{(2)}$ to a fast-update layer, enabling rapid consensus formation and more coherent multimodal integration. Finally, to leverage persistent yet informative signals, CCMS assigns $\delta^{(3)}$ to the last fast-update layer, accelerating memory integration for more comprehensive learning.
>
> **Q3: In the ablation experiments, the authors only discussed the case where $f$ is fixed at 1. Other cases should be discussed in more detail.**
>
> **A3:** We thank the reviewer for this suggestion. We would like to clarify that $f=1$ is essentially equivalent to a conventional setting with a uniform update rate. Comparing with this case highlights the superiority of our multi-timescale design in our nested framework.
>
> To address the reviewer’s concern, we will include results with $f = 1, 2, 3, 4, 5$ on the CREMA-D dataset, as shown in the table below. We observe that the multi-timescale MoNet consistently outperforms all fixed-$f$ settings, demonstrating the effectiveness of our Multi-timescale nested framework.
>
> | Baseline                 | Multi-timescale (MoNet) | Fix $f=1$ | Fix $f=2$ | Fix $f=3$ | Fix $f=4$ | Fix $f=5$ |
> |-------------------------|--------------------------|-----------|-----------|-----------|-----------|-----------|
> | Multimodal Performance  | 83.33               | 82.87     | 83.07     | 82.93     | 81.98     | 82.12     |
>
> **Q4: The experimental setup and implementation details seem insufficiently detailed. Do the authors follow existing methods, such as [A]?**
>
> **A4:** We thank the reviewer for the comment. We would like to clarify that, due to space limitations, more comprehensive experimental and implementation details are provided in our Appendix. Specifically, descriptions of the experimental datasets are provided in **Appendix B.1**, while the evaluation metrics are detailed in **Appendix B.2**. In addition, the implementation details and algorithmic pipeline of our MoNet are presented in **Appendix B.3**.
>
> **References:**
>
> [A] Peng, Xiaokang, et al. "Balanced multimodal learning via on-the-fly gradient modulation." Proceedings of the IEEE/CVF conference on computer vision and pattern recognition. 2022.

---

> > ### Author Rebuttal · Reviewer_FxR5 · 2026-04-02
> >
> > Thanks for the author's response, and my concerns have been well addressed. Thus, I recommend that this paper be accepted.

---

### Official Review · Reviewer_Y4CR · 2026-03-02

**Soundness:** 3
**Presentation:** 4
**Significance:** 3
**Originality:** 3
**Overall Recommendation:** 5
**Confidence:** 5

**Summary:**

This paper proposes a novel Multimodal Nested Learning Framework, MoNet, which is the first nested learning method used to multimodal learning task. MoNet consists of two main nested modules: DMSM for slow and stable encoding and decoding of decoupled modal features, and AMCF using CCMS for adaptive fusion to balance multimodal features. Comparative experiments and ablation experiments demonstrate the general effectiveness and superiority of the proposed MoNet.

**Compliance With Llm Reviewing Policy:**

Affirmed.

**Key Questions For Authors:**

My questions mainly focus on some details of the paper's methodology, and I hope the author can provide some explanations or discussions to resolve my consideration in the weakness section.

**Limitations:**

The authors have thoroughly discussed the potential impacts and limitations.

**Strengths And Weaknesses:**

Strengths:
1. This paper is well-motivated, utilizing a novel nested paradigm to specifically address the issues of existing baseline methods.
2. Extensive and comprehensive experiments demonstrate the effectiveness of the method and its potential for nested learning applications.
3. The targeted treatment of the three complementary components in CCMS is highly innovative and reasonable.
4. The paper is easy to follow, with a well-developed, reasonable, and easy-to-understand organization.

Weaknesses:
1. The specific definitions of ‘unimodal contribution’ mentioned in Figure 1 and ‘modal relative influence’ mentioned in Sec. 4.4 seem to be lacking. Are they sharing commonalities?
2. Different nested layers have different optimizers, which seems to make the method more complex and difficult to tune.
3. Why did the authors mention the claim in lines 186-191? What is the approach for directly measuring features, and how does it differ from current approach? This is currently unknown.
4. A clearer experimental setup (e.g., including dataset, method details, etc.) is needed for reproducibility.

---

> ### Author Rebuttal · Authors · 2026-03-30
>
> Thanks for your valuable comments and insightful suggestions. Attached is our point-by-point response.
>
> **Q1: 1) The specific definitions of ‘unimodal contribution’ mentioned in Figure 1 and ‘modal relative influence’ mentioned in Sec. 4.4 seems to be lacking. 2) Are they sharing commonalities?**
>
> **A1:** We thank the reviewer for the comment. **1)** We want to clarify that both concepts have been defined in the paper and appendix. Specifically, unimodal contribution is introduced in **Section 1 (Figure 1)** and further detailed in our **Appendix A.1**. More specifically, it follows prior work [1], measuring how much each modality contributes to the joint classification predictions. In contrast, modal relative influence is defined in **Section 4.4** and further introduced in our **Appendix B.2 3)**, quantifying how each modality affects the optimization process via alignment with error signals at CCMS layers. **2)** Yes, they share the common goal of characterizing modality dominance. However, they differ in perspective: unimodal contribution reflects the outcome (i.e., prediction-level importance), while modal relative influence captures the process (i.e., optimization-level dynamics).
>
> **Q2: Different nested layers have different optimizers, which seems to make the method more complex and difficult to tune.**
>
> **A2:** Thank you for the reviewer’s insightful comment. In fact, different nested levels with different optimizers do not increase complexity and make tuning more difficult. More specifically, nested learning only tries to decouple and decompose traditional methods into different nested levels, without increasing overall complexity. Additionally, using multiple optimizers provides the flexibility to apply tailored optimizers for different parts of the framework, and it does not cause tuning failure if the optimizers are not extensively adjusted. Specifically, it only involves scheduling updates across modules, which is supported by standard training pipelines and does not require manual tuning.
>
> **Q3: 1) Why did the authors mention the claim in lines 186-191? 2) What is the approach for directly measuring features, and 3) how does it differ from the current approach?**
>
> **A3:** We appreciate the reviewer’s comment. **1)** We aim to use this claim to clarify the motivation behind **Equation (2)**. Specifically, modality-specific features are not directly supervised by their inherent informativeness or completeness, but rather by their contribution with respect to the task semantics, thereby enhancing the feature discrimination. **2)** The approach for directly measuring features is usually based on the informativeness and completeness of modalities themselves to evaluate the quality of the features. **3)** The former seeks to encode features that are more discriminative with respect to task semantics, while the latter tends to encourage the learned features to capture richer information in general, including redundant information that may be irrelevant to the task.
>
> **Q4: A clearer experimental setup (e.g., including dataset, method details, etc.) is needed for reproducibility.**
>
> **A4:** We thank the reviewer for the comment. We would like to clarify that, due to space limitations, more comprehensive experimental details are provided in our supplementary Appendix. Specifically, the dataset descriptions are included in **Appendix B.1**. In addition, implementation details and the algorithmic pipeline of our MoNet are provided in **Appendix B.3**. We will release the code after the peer-review process to ensure reproducibility.
>
> **References:**
>
> [1] Peng, Xiaokang, et al. "Balanced multimodal learning via on-the-fly gradient modulation." Proceedings of the IEEE/CVF conference on computer vision and pattern recognition. 2022.

---

### Official Review · Reviewer_oiD5 · 2026-03-10

**Soundness:** 3
**Presentation:** 2
**Significance:** 3
**Originality:** 3
**Overall Recommendation:** 4
**Confidence:** 5

**Summary:**

The paper propose a novel Multimodal Nested Learning Framework (MoNet) to solve the imbalanced multimodal learning task. The authors employ a Decoupled Multimodal Stable Memory block and an Adaptive Multimodal Coordinated Fusion block to decoupling and coordinating multimodal learning. The introduction of nested learning reformulates the monolithic framework into independent sub-processes, resulting in significant performance improvements in the experiments.

**Compliance With Llm Reviewing Policy:**

Affirmed.

**Final Justification:**

This paper proposes propose a novel Multimodal Nested Learning Framework (MoNet) to solve the imbalanced multimodal learning task. The authors employ a Decoupled Multimodal Stable Memory block and an Adaptive Multimodal Coordinated Fusion block to decoupling and coordinating multimodal learning. In the rebuttal phase, the authors addressed the aforementioned weaknesses. Therefore, I recommend a Weak Accept.

**Key Questions For Authors:**

In the rebuttal stage, the authors should address the aforementioned weaknesses by providing a clearer and more in-depth discussion of their proposed nested framework, and giving more comprehensive experimental analysis, which would help strengthen the paper.

**Limitations:**

Yes, the authors have discussed.

**Strengths And Weaknesses:**

Pros:
- The proposed MoNet is sufficiently innovative, especially its decoupling and coordination, which resulted in superior performance gains in experiments. The paper demonstrates the superiority of the method through extensive experiments, particularly across eight different datasets.
- The authors provided a comprehensive review of related work on imbalanced multimodal learning.
- The experiments are quite extensive, particularly the visualization analysis and supplementary material in Figure 9, which are both insightful and interesting.

Cons:
- Nested learning appears to be relatively new work, differing from previous nested concepts. The authors' descriptions of related work and methods are not comprehensive enough and are somewhat difficult to understand.
- It is unclear why different nesting methods use different loss forms. For example, CMS uses L2 loss instead of cross-entropy.
- It's unclear whether MoNet's superiority lies in addressing modal imbalance or general multimodal learning? Specifically, MoNet outperforms existing methods seems to due to its superior multimodal representations rather than its handling of imbalance.
- The ablation study does not fully examine the effectiveness of the update frequency and the decoupling module. The authors should explore more ablation scenarios to ensure the effectiveness of the modules is well validated. For example, f_f, f_s, w/o CCMS but w/ CMS.
- The implementation of MoNet is complex. The proposed multi-level CCMS and nested optimizers are difficult to tune on existing traditional frameworks. More details need to be disclosed.

---

> ### Author Rebuttal · Authors · 2026-03-30
>
> Thank you for your valuable comments and insightful feedback. Attached is our point-by-point response.
>
> **Q1: The authors' descriptions of related work and methods are not comprehensive enough and are somewhat difficult to understand.**
>
> **A1:** We appreciate the reviewer’s feedback regarding clarity. We acknowledge that the current presentation of the related work and method could be improved in terms of readability, especially for readers unfamiliar with nested learning. In the revision, we will restructure the related work section to provide a clearer categorization of existing approaches and improve the overall narrative flow. For the method section, we have introduced and modeled nested and multimodal learning in **Section 3.1**. We will provide more comprehensive technical details and improve the explanation of the framework to make it more accessible.
>
> **Q2: It is unclear why different nesting methods use different loss forms. For example, CMS uses L2 loss instead of cross-entropy.**
>
> **A2:** We thank the reviewer for this insightful comment. The use of different loss functions is intentional and reflects the distinct objectives of different nested levels. The outer level focuses on **multimodal classification** and therefore adopts cross-entropy loss, while the inner level is designed to **correct optimization signals and operates** in a regression setting, for which L2 loss is more appropriate. This design follows the principle of nested learning, where each level optimizes its own objective. We will clarify it in the revised version.
>
> **Q3: It's unclear whether MoNet's superiority lies in addressing modal imbalance or general multimodal learning? Specifically, MoNet outperforms existing methods, which seems to be due to its superior multimodal representations rather than its handling of imbalance.**
>
> **A3:** We thank the reviewer for this insightful question. We would like to clarify that in all experiments, we fix the backbones across methods, ensuring comparable multimodal representation capacity. Under this controlled setting, the performance gains of MoNet cannot be attributed to stronger general representations but instead reflect its effectiveness in addressing modality imbalance. Specifically, MoNet improves multimodal optimization by decoupling modality-wise learning (DMSM) and coordinating multimodal fusion (AMCF), mitigating gradient interference and adaptively rebalancing strong and lazy modalities. This is further supported by the ablation results in **Table 3**, where ablating these components leads to significant performance degradation, highlighting that the gains primarily stem from improved modality imbalance handling rather than differences in representation capacity.
>
>
> **Q4: The ablation study does not fully examine the effectiveness of the update frequency and the decoupling module. The authors should explore more ablation scenarios. For example, f_f, f_s, w/o CCMS but w/ CMS.**
>
> **A4:** We appreciate the reviewer’s suggestion. We would like to clarify that the update frequencies $f_f$ and $f_s$ are not removable parameters. We have set both of them to 1, ablating the model to a conventional single time-scale baseline, with the results in the **4th row of Table 3**. In addition, the effectiveness of $f_f$ and $f_s$, which are controlled by $\gamma$ in CCMS, has already been analyzed in **Section 4.3** and **Figure 3 (b)**. To address your concern, we will present the results more clearly in the table below.
>
> | $\gamma$      | 1     | 2     | 3     | 4     |
> |--------|-------|-------|-------|-------|
> | Fusion | 80.11 | **82.66** | 81.45 | 81.05 |
> | Audio  | 61.15 | 61.48 | 59.95 | 60.62 |
> | Video  | 74.73 | 75.00 | 73.66 | 72.58 |
>
> Moreover, we would like to clarify that we have already conducted an ablation study for the *w/o CCMS but w/ CMS* setting, as shown in the **7th row of Table 3**. We will refine the presentation of the ablation settings in the next version to avoid misunderstanding.
>
> **Q5: The implementation of MoNet is complex. The proposed multi-level CCMS and nested optimizers are difficult to tune on existing traditional frameworks. More details need to be disclosed.**
>
> **A5:** We thank the reviewer for the concern regarding implementation complexity and tuning. In practice, the proposed CCMS and nested optimizers can be implemented using standard deep learning frameworks without substantial difficulty. Specifically, the CCMS is structurally composed of linearly stacked MLPs and introduces only a small number of hyperparameters (i.e., $\beta_1$ and $\beta_2$), which are not highly sensitive and can be set to common default values as shown in the parameter analysis (**Section 4.3**). The use of multiple optimizers primarily involves scheduling updates across modules, which is supported by standard training pipelines and does not require manual tuning. We will release the code along with detailed documentation to show more details and facilitate reproducibility.

---

> > ### Author Rebuttal · Reviewer_oiD5 · 2026-04-03
> >
> > I thank the authors for their responses. All my concerns have been addressed,and thus I will retain my original score.

---

### Official Review · Reviewer_5XQd · 2026-03-11

**Soundness:** 3
**Presentation:** 3
**Significance:** 4
**Originality:** 3
**Overall Recommendation:** 5
**Confidence:** 4

**Summary:**

This paper proposed a Multimodal Nested Learning Framework, MoNet, which decouples and coordinates multimodal optimization to address imbalanced learning across modalities. The method consists of two main nested components: a Decoupled Multimodal Stable Memory (DMSM) block for independent modality learning, and an Adaptive Multimodal Coordinated Fusion (AMCF) block for balanced multimodal fusion. Extensive experiments demonstrate the multimodal learning superiority of our MoNet on 8 datasets.

**Compliance With Llm Reviewing Policy:**

Affirmed.

**Final Justification:**

After carefully reading the paper and the author's response, I think that the overall score of this paper should be 5.

**Key Questions For Authors:**

1. Which modality-specific way does CCMS address modality imbalance?
2. The unimodal performance lags behind in some cases.
3. Why is there inconsistency between different modalities in imbalance in Tab. 2?

**Limitations:**

Yes

**Strengths And Weaknesses:**

Strengths:
1. The description of the method is detailed and clear, with illustrations of individual components.
2. MoNet is validated on various multimodal benchmarks, including three tasks.
3. ERIN achieves significant multimodal classification performance gains.

Weaknesses:
1. The authors do not clarify whether CCMS explicitly handles different modalities as in existing methods [1-2]. Need to further discuss its specificity.
2. The meaning of the unimodal performance metric in the experiments is unclear, and the proposed method underperforms the baselines in some datasets and modalities.
3. The strong and lazy modalities are inconsistent within the same task (see Tab. 2, 2D–3D dataset).
4. There are some minor notation errors.

[1] Huang, Chengxiang, et al. "Adaptive unimodal regulation for balanced multimodal information acquisition." Proceedings of the Computer Vision and Pattern Recognition Conference. 2025.

[2] Wei, Shicai, Chunbo Luo, and Yang Luo. "Boosting multimodal learning via disentangled gradient learning." Proceedings of the IEEE/CVF International Conference on Computer Vision. 2025.

---

> ### Author Rebuttal · Authors · 2026-03-30
>
> Thanks for your valuable comments and insightful suggestions. We have carefully looked into all the comments and suggestions. Attached is our point-by-point response.
>
> **Q1: The authors do not clarify whether CCMS explicitly handles different modalities as in existing methods [1-2].**
>
> **A1:** Thank you for the insightful comment. CCMS does not explicitly model or reweight different modalities as in prior works [1–2]. Instead, as described in **Lines 223–250 of Section 3.3**, it implicitly captures modality-specific dynamics by decomposing the multimodal optimization signal into multiple temporal CCMS layers.
>
> Unlike existing methods that rely on explicit modality-wise weighting or gradient modulation, CCMS operates directly on the temporal variation of the optimization signal during training. Specifically, we observe that fast-changing components are typically dominated by strong modalities, while slow-changing components are associated with informative modalities. By separating these signals across temporal layers, CCMS can adaptively regulate modality contributions during training, without introducing any explicit balancing mechanism. This design enables CCMS to handle modality imbalance in a data-driven and optimization-centric manner, rather than through manually designed modality-specific operations.
>
> **Q2: 1) The meaning of the unimodal performance metric in the experiments is unclear. 2) The proposed method underperforms the baselines in some datasets and modalities.**
>
> **A2:** **1)** We thank the reviewer for pointing out the ambiguity in unimodal metrics. In our paper, unimodal performance refers to the classification accuracy when only a single modality is used as input, described in detail in **Appendix B.2 2)**. **2)** We acknowledge that our MoNet underperforms several baselines on some unimodal settings. This is primarily because **MoNet is designed to optimize multimodal joint learning rather than unimodal specialization**. Specifically, methods that focus on maximizing the performance of a single dominant modality can achieve higher unimodal accuracy, but this typically leads to over-reliance on that modality, which may hinder balanced multimodal learning. In contrast, MoNet encourages more balanced optimization across modalities by decomposing training signals and mitigating modality dominance. As a result, while this design may slightly limit peak unimodal performance in some cases, it leads to more coordinated multimodal representations and consistently superior multimodal recognition performance across eight datasets.
>
> **Q3: The strong and lazy modalities are inconsistent within the same task (Tab. 2, 2D–3D dataset).**
>
> **A3:** We appreciate this insightful observation. We would like to clarify that **modality dominance is not a fixed property but depends on both dataset characteristics and training dynamics**. As observed and analyzed in **Section 4.4 and Figure 5**, the relative influence of each modality evolves during training, not to mention that it varies with the quality and characteristics of different multimodal data. Therefore, the notions of *strong* and *lazy* modalities are inherently relative and context-dependent.
>
> **Q4: There are some minor notation errors.**
>
> **A4:** We thank the reviewer and will carefully proofread and fix all notation inconsistencies.
>
> **References:**
>
> [1] Huang, Chengxiang, et al. "Adaptive unimodal regulation for balanced multimodal information acquisition." Proceedings of the Computer Vision and Pattern Recognition Conference. 2025.
>
> [2] Wei, Shicai, Chunbo Luo, and Yang Luo. "Boosting multimodal learning via disentangled gradient learning." Proceedings of the IEEE/CVF International Conference on Computer Vision. 2025.

---

> > ### Author Rebuttal · Reviewer_5XQd · 2026-04-02
> >
> > The authors have addressed all my concerns well. So, I think that this paper can be accepted.

---

### Decision · Program_Chairs · 2026-04-30

**Decision:**

Accept (spotlight)

**Comment:**

This paper proposes MoNet, a novel multimodal nested learning framework that decouples and coordinates multimodal optimization to address imbalanced learning across modalities. The authors conducted extensive experimental validation across eight datasets and three tasks, and achieved significant performance gains. The reviewers acknowledge the method's strong technical soundness. While concerns were raised regarding presentation clarity, ablation completeness, and implementation complexity, the authors have adequately addressed these issues during the rebuttal phase. Overall, the paper presents a technically solid contribution with high impact on multimodal learning. Based on the unanimous positive recommendations after rebuttal,  the paper is recommended for acceptance.